



# The role of grain-size evolution on the rheology of ice: Implications for reconciling laboratory creep data and the Glen flow law

Mark D. Behn[1] David L. Goldsby[2], Greg Hirth[3]

[1]Dept. Earth & Environmental Sciences, Boston College, Chestnut Hill, MA  02467 USA
[2]Dept. Earth & Environmental Science, University of Pennsylvania, Philadelphia, PA  19104 USA
[3]Dept. Earth, Environmental & Planetary Sciences, Brown University, Providence, RI  02912 USA

*Correspondence to*: Mark D. Behn (mark.behn@bc.edu)

**Abstract.** Viscous flow in ice is often described by the Glen flow law—a non-Newtonian, power-law relationship between stress and strain-rate with a stress exponent $n \sim 3$.  The Glen law is attributed to grain-size-insensitive dislocation creep; however, laboratory and field studies demonstrate that deformation in ice can be strongly dependent on grain size.  This has led to the hypothesis that at sufficiently low stresses, ice flow is controlled by grain boundary sliding, which explicitly incorporates the grain-size dependence of ice rheology.  Experimental studies find that neither dislocation creep ($n \sim 4$) nor grain boundary sliding ($n \sim 1.8$) have stress exponents that match the value of $n \sim 3$ in the Glen law.  Thus, although the Glen law provides an approximate description of ice flow in glaciers and ice sheets, its functional form is not explained by a single deformation mechanism.  Here we seek to understand the origin of the $n \sim 3$ dependence of the Glen law by using the "wattmeter" to model grain-size evolution in ice.  The wattmeter posits that grain size is controlled by a balance between the mechanical work required for grain growth and dynamic grain size reduction.  Using the wattmeter, we calculate grain size evolution in two end-member cases: (1) a 1-D shear zone, and (2) as a function of depth within an ice-sheet.  Calculated grain sizes match both laboratory data and ice core observations for the interior of ice sheets.  Finally, we show that variations in grain size with deformation conditions result in an effective stress exponent intermediate between grain boundary sliding and dislocation creep, which is consistent with a value of $n = 3 \pm 0.5$ over the range of strain rates found in most natural systems.

## 1  Introduction

Glaciers and ice sheets deform via gravity-driven viscous flow. The most widely employed constitutive description of ice flow is the grain-size independent Glen law, a power-law expression between strain rate ($\dot{\varepsilon}$) and stress ($\sigma$) of the form $\dot{\varepsilon} = B\sigma^n$, where $B$ is a temperature-dependent constant that embodies the Arrhenius dependence of creep. The Glen law is characterized by a stress exponent $n$ of ~3, and is based on the classic laboratory experiments of Glen (1952; 1955) and numerous subsequent experiments on coarse-grained polycrystalline ice. Applications of the Glen law to natural settings have found that it provides a good description of flow in glaciers and ice sheets (e.g., Weertman, 1983). For example, it has been shown that the flow-line morphology of the Greenland and west Antarctic ice sheets (Cuffey, 2006), as well as smaller Antarctic ice caps (Martin



& Sanderson, 1980; Hamley et al., 1985; Young et al., 1989), are consistent with a stress exponent of ~3. Further, the relationship between stress and strain rate in spreading ice shelves (Thomas, 1973; Jezek et al., 1985), as well as borehole tilt measurements in temperate glaciers (Raymond, 1973; 1980) and ice sheets (Paterson, 1983) also support the lab-derived value of $n \sim 3$.

Yet despite the Glen law's widespread adoption in ice-flow models, several lines of evidence indicate that it is an oversimplification of the rheological behavior of ice. Observational data sets indicate that grain size, fabric development, impurities, and water content can all influence the creep behavior of ice (Cuffey & Paterson, 2010). These effects are often parameterized with an enhancement factor, which modifies the $B$ term in the Glen law. In particular, grain size variations have been shown to influence creep rates in basal ice in cores from Greenland and Antarctica (e.g., Cuffey et al., 2000).

From the laboratory perspective, the Glen law fails to describe ice rheology over a wide range of stresses (Pimienta et al., 1987; Duval & Castelnau, 1995; Durham & Stern, 2001; Goldsby & Kohlstedt, 2001; Montagnat & Duval, 2004), with an observed stress exponent $n > 3$ at high stress and $n < 3$ at low stress (Fig. 1). Indeed, Glen (1952) originally determined a value of $n = 4$ based on early experimental data at stresses of 0.2–1 MPa. The low-$n$ regime suggested by more recent laboratory data for samples of comparatively coarse grains sizes (~0.1 mm) is of particular importance for glaciology because

it indicates a potential transition to a low-$n$ creep mechanism at typical glacier stresses ($\leq \sim 0.1$ MPa). Values of $n \sim 2$ are often associated with creep mechanisms that involve dislocation-accommodated grain boundary sliding (GBS), which are strongly dependent on grain size. Mechanisms involving GBS are characterized by increasing strain rate with decreasing grain size, i.e., $\dot{\varepsilon} \propto d^{-m}$, where $d$ is grain size and the grain size exponent $m$ has a value of 1–3 depending on the mechanisms that accommodate GBS creep (e.g., Poirier, 1985; Langdon, 1994).

Most early laboratory experiments on ice, such as those by Glen (1952; 1955), focused on polycrystalline samples with grain sizes typical of natural settings (1–10 mm). However, these data are difficult to interpret in terms of a GBS creep mechanism at low stresses because it is hard to separate to steady-state from transient creep (Weertman, 1983). To access low-stress (low-$n$) creep mechanisms on a practical timeframe requires fabrication of specimens with grain sizes that are much smaller than typically found in terrestrial ice (Goldsby & Kohlstedt, 2001; Durham et al., 2001). Creep experiments on such

samples reveal a stress exponent of $n = 4$ at high stresses with no grain size dependence and are interpreted to reflect a dislocation creep mechanism (Goldsby & Kohlstedt, 2001). By contrast, with decreasing stress the data reveal the existence of a creep regime characterized by $n = 1.8$ (Fig. 1) and a marked dependence on grain size with $m = 1.4$. These values of $n$ and $m$ are consistent with a GBS creep mechanism in which GBS is accommodated by dislocation motion (Nieh et al., 1997).

These laboratory data lead to a paradox for interpreting the behavior of ice flow in natural settings—namely, neither the

laboratory-derived stress exponents for dislocation creep ($n \sim 4$), nor for dislocation-accommodated GBS ($n \sim 1.8$), match the value of $n \sim 3$ in the Glen law. One possible explanation for this discrepancy is that variations in ice grain size will influence the relative contributions of GBS and dislocation creep, leading to a transitional regime between these two creep mechanisms (Peltier et al., 2000; Goldsby, 2006). To evaluate this hypothesis, it is necessary to quantify how grain size evolves spatially





and temporally within glaciers and ice sheets. A number of studies have investigated the competing effects of grain growth
and dynamic recrystallization on grain size in ice (e.g., Alley, 1992; Alley et al., 1995; Duval & Castelnau, 1995; De La
Chapelle et al., 1998; Montagnat & Duval 2000; Durand et al., 2006; Roessiger et al., 2011; Ng & Jacka, 2017). Faria et al.
(2014a) proposed a fully coupled model in which steady-state grain size is described as a function of temperature and strain-
rate, but in deriving an expression for steady-state grain size they assumed the grain size independent Glen law. Here, we
develop a unified description of grain size and deformation that explicitly accounts for the experimental constraints on grain
size sensitive creep.

We build on the framework for grain size evolution proposed by Faria et al. (2014a). We do so by adapting the "wattmeter"
(Austin & Evans, 2007; 2009), originally developed to quantify grain size evolution in crustal and mantle rocks, to calculate
grain sizes in ice. The wattmeter is based on the concept that grain size in any solid crystal aggregate is controlled by the
balance of the mechanical work required for grain growth and dynamic recrystallization. Coupling the wattmeter with a
composite flow law that incorporates both GBS and dislocation creep, we (1) develop a model that provides a self-consistent
description of deformation and grain size evolution in ice, and (2) test our model using constraints from laboratory data and
natural settings. Lastly, we show that grain size evolution in response to deformation leads to an effective stress exponent that
is intermediate between grain boundary sliding and dislocation creep, consistent with the $n \sim 3$ value of the Glen law.

## 2 Grain-size evolution model for ice

Several models have been proposed to quantify the evolution of grain size in pure ice. The simplest of these models is the
piezometric relationship, in which grain size is related directly to the inverse of stress (e.g., Azuma & Higashi, 1983; Jacka &
Jun, 1994). However, the piezometer does not account for the physical processes that control ice grain size—namely the
competition between grain growth and grain-size reduction via recrystallization (e.g., Alley, 1992). Near the surface, ice core
data show a monotonic increase in grain size with depth, indicating that grain growth is the dominant process controlling grain
size (Gow et al., 1997). However, at greater depths, grain sizes often stabilize, suggesting a steady-state in which the rate of
recrystallization balances the rate of grain growth (e.g., Roessinger et al., 2011; Faria et al., 2014b). Similar processes are
thought to occur in crustal and mantle rocks and have led to models that either assume grain growth and recrystallization are
balanced at the field boundary between GBS and dislocation creep (de Bresser et al., 2001) or that explicitly calculate the rates
of grain growth and grain size reduction (e.g., Hall & Parmentier, 2003; Montési & Hirth, 2003; Bercovici & Ricard, 2014).
A particularly successful model, which accurately predicts grain sizes in a range of natural samples (e.g., calcite, quartz,
olivine), is the "wattmeter" (Austin & Evans, 2007; 2009). The wattmeter posits that the mean grain size, $d$, of a volume of
rock or ice is controlled by the balance of mechanical work required for grain growth and dynamic recrystallization.
Specifically, the wattmeter calculates the rate of grain-size evolution from the competing rates of grain growth and dynamic
recrystallization:





$$\dot{d} = \dot{d}_{gg} - \dot{d}_{red} \qquad (1)$$

where $\dot{d}$ is the change in mean grain size with respect to time, $\dot{d}_{gg}$ is the rate of grain growth, and $\dot{d}_{red}$ is the rate of grain size reduction or "polygonization" (Alley et al., 1995). Below we describe our approach for calculating the rates of grain growth and grain size reduction and how the grain size evolution law in Eq. (1) can be coupled with a composite flow law that includes both GBS and dislocation creep to predict the effective stress exponent for ice.

## 2.1 Grain growth

Following Alley et al. (1986) we assume that grain growth can be described by a relationship of the form:

$$d^p - d_o^p = Kt \qquad (2)$$

where $K$ follows an Arrhenius relation:

$$K = K_{gg} \exp\left(-\frac{Q_{gg}}{RT}\right) \qquad (3)$$

In these equations $d_o$ is an initial grain size, $p$ is the grain growth exponent, $t$ is time, $K_{gg}$ is the grain growth constant, $Q_{gg}$ is the activation enthalpy for grain growth, $R$ is the universal gas constant, and $T$ is temperature. Substituting Eq. (3) into Eq. (2) and differentiating with respect to time allows us to write an expression for the rate of grain growth:

$$\dot{d}_{gg} = p^{-1} d^{1-p} K_{gg} \exp\left(-\frac{Q_{gg}}{RT}\right) \qquad (4)$$

Equation (4) provides a general expression for grain growth; however, the values of the grain growth parameters, $p$, $K_{gg}$, and
$Q_{gg}$ are not well constrained in natural systems and depend on the presence of microparticles, bubbles, and/or other impurities in the ice (e.g., Alley et al., 1986). In Sect. 2.5 we will describe our approach for estimating these parameters using a combination of laboratory and ice core data.

## 2.2 Grain size reduction

The wattmeter posits that the rate of grain size reduction $\dot{d}_{red}$ is controlled by the rate of mechanical work and the rate at
which this work is dissipated (Austin & Evans, 2007; 2009; Bercovici & Ricard, 2012). Specifically, the rate of mechanical work per unit volume, $\dot{W}_{tot}$, is defined as:



$$\dot{W}_{tot} = \sigma\dot{\varepsilon} \tag{5}$$

where $\sigma$ is stress and $\dot{\varepsilon}$ is strain rate (assuming that the rate of stress change is negligible over the timescale of grain size evolution). This work rate must be balanced by the rate at which the internal energy of the system, $\dot{E}_{int}$, increases plus the rate at which energy is dissipated, $\dot{\theta}_{irr}$:

$$\dot{W}_{tot} = \sigma\dot{\varepsilon} = \dot{E}_{int} + \dot{\theta}_{irr} \tag{6}$$

The increase in internal energy can be related to the increase in grain boundary area:

$$\dot{E}_{int} = \frac{-c\gamma}{d^2}\dot{d}_{red} \tag{7}$$

where $\gamma$ is the grain boundary energy and $c$ is a geometrical factor ($=\pi$ for spherical grains). The rate of dissipation in Eq. (6) is related to the fraction, $\lambda$, of the total work rate that is responsible for increases in internal energy:

$$\dot{\theta}_{irr} = (1-\lambda)\sigma\dot{\varepsilon} \tag{8}$$

Here we note a difference in the application of the wattmeter to ice as compared to crustal and mantle rocks. In most terrestrial minerals, the two primary creep mechanisms are diffusion and dislocation creep. Because grain growth during diffusion creep was shown to be the same as that during static conditions (Karato et al., 1986), the work done by diffusion creep is assumed to be completely dissipated (i.e., $\lambda_{diff} = 0$) and only dislocation creep leads to grain size reduction. By contrast, under Earth-like pressure and temperature conditions, ice deformation proceeds primarily by a combination of GBS and dislocation creep (Goldsby & Kohlstedt, 2001). Because some fraction of the work done by both GBS and dislocation creep will lead to grain size reduction (i.e., $\lambda_{disl}$ & $\lambda_{GBS} > 0$) the dissipation rate can be re-written as:

$$\dot{\theta}_{irr} = (1-\beta)(1-\lambda_{GBS})\sigma\dot{\varepsilon} + \beta(1-\lambda_{disl})\sigma\dot{\varepsilon} \tag{9}$$

where

$$\beta = \frac{\dot{W}_{disl}}{\dot{W}_{tot}} \tag{10}$$

Here we assume that the total work rate can be expressed as the sum of the contributions from the individual deformation mechanisms:





$$\dot{W}_{tot} = \dot{W}_{disl} + \dot{W}_{GBS} = \sigma\dot{\varepsilon}_{disl} + \sigma\dot{\varepsilon}_{GBS} \tag{11}$$

The values of $\lambda_{disl}$ and $\lambda_{GBS}$ are uncertain, but following Austin & Evans (2007; 2009) we investigate values for both

parameters in the range of 0.005–0.05.

Substituting Eqs. (7) and (9) into Eq. (6) we derive an expression relating the rate of grain size reduction to the total work rate:

$$\dot{d}_{red} = \frac{\left(\lambda_{GBS} - \beta\lambda_{GBS} + \beta\lambda_{disl}\right)d^2}{-c\gamma}\sigma\dot{\varepsilon} \tag{12}$$

The final grain size evolution equation can then be assembled from Eqs. (1), (4) and (12):

$$\dot{d}_{tot} = p^{-1}d^{1-p}K_{gg}\exp\left(-\frac{Q_{gg}}{RT}\right) - \frac{\left(\lambda_{GBS} - \beta\lambda_{GBS} + \beta\lambda_{disl}\right)d^2}{-c\gamma}\sigma\dot{\varepsilon} \tag{13}$$

It is often useful to define a steady-state grain size, $d_{ss}$, which occurs when $\dot{d}_{tot} = 0$ :

$$d_{ss}^{1+p} = \frac{K_{gg}\exp\left(-\frac{Q_{gg}}{RT}\right)p^{-1}c\gamma}{\left(\lambda_{GBS} - \beta\lambda_{GBS} + \beta\lambda_{disl}\right)\sigma\dot{\varepsilon}} \tag{14}$$

The concept of a steady-state grain size is analogous to that derived by Faria et al. (2014a), with the exception that they assumed

creep was governed exclusively by the Glen law and grain size was related to stress-only (e.g., Jacka & Jun, 1994) rather than

to the work rate (Eq. 6). In practice the steady-state grain size may not be achieved if there is insufficient time for grains to

fully evolve to be in equilibrium with the surrounding deformation conditions. In these situations, Eq. (13) must be solved

and coupled with the governing equations and constitutive relationships.

**2.3 Composite rheology for ice**

To apply the grain size evolution model defined by Eq. (13) to natural systems we calculate the relative rates of deformation

by GBS and dislocation creep. To do so, we formulate a two-mechanism composite flow law that contains additive

contributions from each creep mechanism of the form:

$$\dot{\varepsilon} = \dot{\varepsilon}_{GBS} + \dot{\varepsilon}_{disl} \tag{15}$$



This composite law has been used to model the rheology of ice satellites (Barr & McKinnon, 2007) and the relative contribution of GBS and dislocation creep in ice sheets (Kuiper et al., 2020). Here the creep mechanisms are assumed to be independent

and each term on the right-hand-side of Eq. (15) is expressed as a flow law of the general form:

$$\dot{\varepsilon}_i = A_i d^{-m_i} \sigma^{n_i} \exp\left(-\frac{Q_i}{RT}\right) \tag{16}$$

where $A_i$ is a material constant, $m_i$ is the grain size exponent for creep, $n_i$ is the stress exponent, and $Q_i$ is the activation energy. The subscript $i$ denotes the parameters that depend on the deformation mechanism (e.g., GBS or dislocation creep). We note that Goldsby & Kohlstedt (2001) presented a more complicated composite law that includes a term for creep limited by basal

dislocation slip and also a theoretical flow law for diffusion creep. However, extrapolations to grain sizes typical of glaciers and ice sheets demonstrates that neither of these additional creep mechanisms are likely to be important for the flow of terrestrial ice bodies. A list of flow law parameters required to extrapolate Eq. (16) to the full temperature range (up to the melting point) is given in Goldsby & Kohlstedt (2001).

### 2.4 Model setup

To solve for grain size evolution in ice we consider two scenarios: (1) deformation in a shear zone under an imposed velocity contrast $V_{sz}$, and (2) deformation in 1-D vertical column of ice with an assumed surface slope, $\alpha$. For the case of a shear zone with no along-strike pressure and/or viscosity gradients, the shear stress, $\tau$, will be constant and a function of only the viscosity and velocity contrast:

$$\tau = \eta \frac{\partial v}{\partial w} \tag{17}$$

where $\eta$ is the viscosity, $v$ is the velocity parallel to the shear zone, and $w$ is the direction perpendicular to the strike of the shear zone. Integrating Eq. (17) over the width of the shear zone, $w_o$, allows us to write stress in terms of the imposed velocity:

$$\tau = V_{sz} \left(\int_o^{w_o} \frac{1}{\eta} dw\right)^{-1} \tag{18}$$

In the case of deformation within a column of ice with a zero-slip basal boundary condition, the shear stress can be calculated from the surface slope and increases linearly as a function of depth, $z$, in the ice sheet:

$$\tau(z) = \rho g (H - z) \sin(\alpha) \tag{19}$$

Here $\rho$ is the density of ice, $g$ is the gravitational acceleration, and $H$ is the thickness of the ice sheet.



## 2.5 Calibration of grain growth parameters

Before using the wattmeter to predict grain sizes in natural systems, we must first constrain the grain growth parameters used in the model as they will directly control the balance between grain growth and grain size reduction. As noted above grain growth rates in ice are highly sensitive to the presence of impurities, both soluble (e.g., bubbles, ions) and insoluble (e.g., dust/microparticles) (Alley et al., 1986). While the expressions for grain size evolution derived above do not explicitly account for the effects of impurities, we can parameterize their effects through their influence on grain growth. To constrain the grain growth parameters ($p$, $K_{gg}$, and $Q_{gg}$) in Eq. (4) we turn to a combination of laboratory and ice core data. Azuma et al. (2012) measured grain growth rates in laboratory samples both with and without bubbles and found that the grain growth exponent for bubble-free ice was relatively small ($p \sim 2$), but was significantly larger ($p \sim 7$–9) in ice containing bubbles (Fig. 2). The increase in the grain growth exponent in the presence of bubbles was interpreted to reflect the role of "impurity drag".

To investigate the applicability of these experimentally-derived grain-growth rates to natural systems, we compared them to grain sizes in the shallow portions of the GRIP and GISP2 ice cores where recrystallization rates are expected to be small and the increase in grain size with depth dominantly reflects the rate of grain growth (Gow et al., 1997). We use only grain sizes from the depth range between 150 m (~500 yr; taken to represent the depth at which the ice is fully compacted), and 300 m (~1500 yr; below which grain sizes no longer increase at a constant rate, indicative of the influence of recrystallization). For comparison with the laboratory data, depth was converted to time for the GRIP and GISP2 cores based on the age models of Dansgaard et al. (1993) and Ram et al. (2000), respectively.

Using experiments conducted at the temperature conditions found between 150–300 m depth in the GRIP and GISP2 ice cores (243 K; Hvidberg et al., 1997), we first refit the Azuma et al. (2012) experimental data for the grain growth parameters $p$ and $K_{gg}$ using the approach of Bons et al. (2001). We find grain growth exponents in the range of 7.1–8.4 for experiments with bubbles and $p = 1.8$ for the single experiment without bubbles (red & blue curves in Fig. 2). Extrapolating these parameters to time-scales applicable to glaciers and ice sheets (e.g., $10^3$–$10^5$ yr), we show that (1) the grain growth parameters derived for ice with bubbles provide a significantly better fit to the ice core data compared to the grain-growth rates for bubble-free ice (compare red vs. blue curves in Fig. 2) and (2) the parameters derived from experiment AL5 provide the best overall fit to the ice core data. However, there is some variability in the experimental data—possibly reflecting differences in bubble content and/or the difficulty in extrapolating grain growth parameters determined on time-scales of hours to days in the laboratory to time-scales of thousands of years in natural systems. In an attempt to address these issues, we refit the Azuma et al. (2012) data from all 3 experiments containing bubbles at 243 K (AL5, AM5, & AS5) jointly with the GRIP ice core data. We do not include the GISP2 data in this fit, as we will calculate grain size as a function of depth throughout the entire GISP2 core in Sect. 3.3 below. The joint fit results in a grain-growth exponent $p$ of $6.03 \pm 0.25$ (solid black line in Fig. 2), slightly less than the values derived from the individual laboratory experiments. Below we use both the grain-growth parameters derived exclusively from experiment AL5 and from the joint fit between the experimental and ice core data (Table 1) in our





application of the wattmeter and discuss the influence of the grain-growth exponent on the derived effective stress exponent

for creep in ice.

## 3. Results

As described above we have used the theoretical framework of the wattmeter (Austin & Evans, 2007; 2009) to develop a new grain size evolution model for ice. In the following section, we will apply this grain size evolution model (loosely referred to as the "wattmeter") to estimate grain size in several simplified systems where deformation is driven by either an imposed

velocity contrast across a 1-D shear zone or by a variation in stress with depth associated with a fixed surface slope.

### 3.1 Steady-state grain size in a shear zone

We first use the wattmeter to predict grain size in a steady-state shear zone deforming at a fixed strain-rate. This setup is analogous to constant strain rate laboratory experiments, such as those by Piazolo et al. (2013) discussed in following Section. In this end-member, we calculate steady-state grain size by iterating between Eqs. (14) and (18) and assuming the grain growth

parameters from our joint fit of the Azuma et al. (2012) experiments and the GRIP ice core data. The result is an estimate of stress and grain size within the shear zone for any imposed strain-rate; Figs 3a & 3b show these estimates calculated at temperatures of 240 K and 265 K, respectively. As noted above the dominant deformation mechanism in ice is sensitive to both grain size and stress, with higher stresses and larger grain sizes favoring dislocation creep and lower stresses and smaller grain sizes favoring GBS-limited creep (Fig. 1). We illustrate the transition between dislocation and GBS creep (often referred

to as the "field boundary") using a deformation mechanism map (Fig. 3). Here we assume that a deformation mechanism acting in kinetic parallel with other creep mechanisms is the dominant mechanism if it yields the fastest creep rate. By overlaying the stresses and grain sizes predicted from the wattmeter on deformation maps calculated at the corresponding temperature, we show how variations in strain rate lead to a transition in the dominant deformation mechanism (Fig. 3).

The relationship between grain size and stress predicted by the wattmeter does not change significantly as a function of

temperature, but has steeper slope compared to either the field boundary or the piezometer (Jacka & Jun, 1994). For example, both the 240 K and 265 K shear zones predict a transition from dislocation to GBS-limited creep at a grain size of 0.2–0.3 mm and a stress of 1–2 MPa (Fig 3). By contrast, the strain rate at which the shear zone is predicted to cross the field boundary varies from $3\times10^{-9}$ s$^{-1}$ to $1\times10^{-6}$ s$^{-1}$ for temperatures of 240 K and 265 K, respectively. These results indicate that when grain size is allowed to vary with the evolving deformation conditions, the dominant deformation mechanism will not be strongly

affected by variations in temperature, but the strain rate corresponding to a specific grain size (and stress) will vary due to the Arrhenius behavior of creep (Eq. 16).

We also examine the relationship between stress and strain rate in the shear zone, comparing cases with a fixed grain size to those in which grain size evolves according to the wattmeter (Fig. 4a). Consistent with the laboratory experiments shown in Fig. 1, the fixed grain size calculations show a distinct change in slope corresponding to the transition from a stress exponent





of $n = 1.8$ in the GBS-limited creep regime to a value of $n = 4$ in the dislocation creep regime (Fig. 4b). By contrast, the

wattmeter predicts a more subdued change in slope in the GBS-limited field corresponding to a higher effective stress exponent

(~2.5) than the lab-derived value of $n = 1.8$. At higher strain rates and stresses the wattmeter converges to the dislocation creep

stress exponent (Fig. 4b). We discuss the origin of these differences in the effective stress exponent in Sect. 4.1.

### 3.2 Application of the shear zone model to laboratory experiments

Piazolo et al. (2013) investigated grain size changes as a function of strain in a series of experiments conducted at different

strain rates. These experiments are ideal for benchmarking and calibrating the wattmeter as we can compare the final grain

size to the steady state value in Eq. (14) and also evaluate the evolution of grain size as a function of time (determined from

the strain given an imposed strain rate) using Eq. (13). Here we investigate a series of cases using the grain growth parameters

from the joint fit of the Azuma et al. (2012) experiments and the GRIP ice core data, as well as those derived exclusively from

experiment AL5 (Fig. 5). Further, we vary the fraction of the total work rate that is responsible for increases in internal energy

assuming for simplicity that $\lambda = \lambda_{GBS} = \lambda_{disl}$. Following the experimental setup of Piazolo et al. (2013), we assume an initial

grain size of 0.5 mm, and use Eq. (13) to calculate grain size as a function of strain for the strain rates used in the experiments.

Simulations were performed to a strain of 0.2, by which time all cases have achieved a steady-state grain size.

        As expected, increasing $\lambda$ results in a smaller steady-state grain size and a more rapid convergence to the steady-state

value with increasing strain (Fig. 5). In general, all cases produce the relative variations in grain size as a function of strain-

rate shown by the experimental data; however, the grain growth parameters derived from experiment AL5 combined with

$\lambda = 0.005 - 0.01$ provide better fits to the data (Figs. 5e,f).

### 3.3 Steady-state grain size in a 1-D vertical column of ice

        We next investigate predictions of the wattmeter for a 1-D vertical column of ice in which stress as a function of depth is

controlled by the surface slope and ice density (Eq. 19). This setup is analogous to deformation within a deforming ice body

and thus can be directly compared with grain size values derived from ice cores. We first simulate a theoretical 1-km column

of ice with a surface slope of 3º, ice density of 930 kg/m³, and a constant temperature of 253 K. We calculate the steady-state

grain size, velocity, strain rate, and effective stress exponent as function of depth assuming $\lambda_{GBS} = \lambda_{disl} = 0.01$ (Fig. 6a).

The effective stress exponent is calculated from the numerical solution using the in local gradient in stress and strain rate with

depth. Grain size decreases with depth due to the increase in stress lower in the column (which drives recrystallization), while

grain growth dominates near the surface. Compared to cases with a constant grain size, grain size evolution produces larger

gradients in velocity and strain rate with depth as the fine grained ice softens near the bed (Fig. 6b,c). Calculations with grain

growth parameters derived from either the joint fit of the experimental and ice core data, or those derived exclusively from





experiment AL5 result in very similar grain size profiles, with the joint fit predicting slightly smaller grain sizes and
correspondingly higher strain rates at the base of the column.

The profiles of velocity and strain-rate have a similar functional form to those calculated for a fixed grain size; however, the effective stress exponent varies significantly between the fixed grain size cases and those with grain size evolution. With a fixed grain size, the effective stress exponent varies from $n_{eff} = 1.8$ at the surface, to $n_{eff} = 2.6$ to $3.7$ at the bed for grain sizes of 1 and 10 mm (Fig. 6d). By contrast, the wattmeter predicts an effective stress exponent that varies from ~2.5 at the surface to ~3 at the bed. Thus, similar to the fixed-width shear zone models, the 1-D vertical column predicts effective stress exponents more similar to the Glen law value compared to cases with a fixed grain size.

Finally, we compare the wattmeter predictions to those using a piezometric relationship relating grain size directly to stress (Jacka & Jun, 1994). The piezometer predicts significantly larger grain sizes in the shallow portion of the column compared to the wattmeter, but reaches similar values near the bed (green curves in Fig. 6). Overall the piezometer results in smaller strain-rates throughout most of the column and a significantly higher effective stress exponent ($n_{eff} \sim 3.9$), similar to the experimental value for dislocation creep.

## 3.4 Application of 1-D ice column model to ice core data

To investigate how well the wattmeter predicts grain sizes observed in natural ice cores, we next apply the 1-D vertical column model to grain sizes measured in the GISP2 ice core (Gow et al., 1997) using the linear intercept method (Alley & Woods, 1996). For comparison to GISP2, we assume a column thickness of 3 km and the temperature profile of Clow et al. (1999), which varies from ~241 K at the surface to 263 K at the bed. Stress is calculated using a constant ice density of 930 kg/m$^3$ (Gow et al., 1997) and a surface slope of 0.11° (Hvidberg et al., 1997). We assume $\lambda_{GBS} = \lambda_{disl} = 0.01$ given the success in using these values to reproduce the Piazolo et al. (2013) experimental data.

One important caveat of the 1-D column models shown in Fig. 6 is that the time-scale to reach a steady-state grain size, particularly in the shallow portion of the column where strain-rates are small, may be greater than $10^{4-5}$ yr. Thus, to compare our model predictions with the ice core data, where the shallowest ice is the youngest ice, we use the time-dependent formulation in Eq. (13) and calculate grain size as a function of time at each depth assuming a fixed surface slope. The age of the ice at each depth is taken from Ram et al. (2000). Incorporating time dependence into our 1-D column calculations does not change the predicted grain sizes near the base of the column where the ice is sufficiently old for grain size to reach steady state. However, it significantly reduces grain sizes in the shallow part of the column, where the young ice does not have sufficient time to reach steady-state (dotted curves, Fig. 7).

Overall, we find a good fit between the grain sizes predicted by the wattmeter and those recorded in the GISP2 ice core. Surface velocities predicted by the wattmeter (~1 m/yr) are also in agreement with those observed near the GISP2 site (Hvidberg et al., 1997). There is little sensitivity to using the grain growth parameters from the Azuma et al. (2012) AL5 experiment only (red curves, Fig. 7) versus the joint fit to all experiments and the ice core data (blue curves, Fig. 7). The one



major deviation between the grain size predictions of the wattmeter and the observed grain sizes occurs at the very base of the core. In this region, observed grain sizes increase up to ~10 mm at the bed, while the wattmeter predicts grain sizes that monotonically decrease to a value of ~2 mm. These deviations are discussed in Sect. 4.2 below.

## 4. Discussion

Grain size is a key microphysical property of ice, controlling not only its creep behavior, but also fracture toughness, melt permeability, and seismic attenuation and wave-speeds. Thus, knowledge of its variability is critical to interpreting the physical properties and dynamic behavior of ice sheets and glaciers. The success of the wattmeter in predicting the grain sizes observed in both the Piazolo et al. (2013) shear zone experiments (Fig. 5) and the GISP2 ice core data (Fig. 7) provides a strong indication that the wattmeter captures the first order physics of grain size evolution in ice. We emphasize that the fit of the model to

these two very different systems is achieved using the same model parameters and require no setting-specific tuning of the model. In the discussion below, we first consider the implications of grain size evolution in reconciling the laboratory creep data with the Glen law. Second, we explore the application of our model to the interpretation of grain size in ice core data. Finally, we discuss the implications of grain size evolution on strain enhancement and strain localization in ice sheets and glaciers.

**4.1 Implications for the Glen law and the stress exponent in ice**

As illustrated in both the steady-state shear zone models (Fig. 4b) and the simulations of a 1-D column of ice deforming due to a surface slope (Fig. 6b), the wattmeter results in an effective stress exponent that is intermediate between the lab-derived values for dislocation and GBS-limited creep and approaches the $n = 3$ value of the Glen law. To interpret these results, we reconsider the end-member cases of deformation accommodated solely by either dislocation or GBS-limited creep. In the

dislocation creep regime, deformation is not sensitive to grain size (i.e., $m_{disl} = 0$ in Eq. 16) and we expect no difference in creep behavior or the effective stress exponent as a function of grain size. By contrast, in GBS-limited creep, strain rate is sensitive to both stress and grain size. Further, the steady-state grain size calculated by the wattmeter will vary as a function of stress and strain-rate (Eq. 14). Thus, substituting the expression for steady-state grain size, $d_{ss}$ in Eq. (14), into the flow law (Eq. 16) we find that strain rate can be related to stress through an effective stress exponent $n_{eff}$ that is proportional to $n_{GBS}$,

$m_{GBS}$, and the grain growth exponent $p$:

$$n_{eff} = \frac{n_{GBS}\left(1 + p\right) + m_{GBS}}{1 + p - m_{GBS}} \tag{20}$$

Using laboratory-determined values for $n_{GBS}$ and $m_{GBS}$ (Table 1) and the grain growth exponent fit by the laboratory and ice core data ($p = 6.2$), we find $n_{eff}$ for GBS-limited creep is equal to $\sim 2.5$. This value corresponds to the effective stress exponent



calculated in the shear zone at low stress and strain-rate (Fig. 4b) and is higher than the laboratory-derived value at a constant

grain size.

We note that this expression for the effective stress exponent is only valid in the limit of steady-state grain size. Processes that limit a change of grain size in the GBS regime will result in a stress exponent closer to the lab-derived value. For example, some experiments have shown that grain growth may be limited during GBS creep (Goldsby & Kohlstedt, 2001; Caswell & Cooper, 2017); moreover, in natural ice impurities may also limit grain growth (e.g., Alley & Woods, 1996). Future

experiments under different conditions (e.g., initial grain size, impurity/bubble distribution, deformation mechanism) are necessary to further constrain these effects on grain growth.

Comparison of the constant grain size shear zone models to those using the wattmeter shows that both predict a transition in the effective stress exponent near the field boundary at strain rates of $10^{-2}$ to $10^{2}$ yr$^{-1}$ (Fig. 4b). When grain size is fixed and does not evolve, $n_{eff}$ varies from the lab-derived values for GBS (1.8) and dislocation creep (4) and only coincides with the

Glen law ($n = 3\pm0.5$) over a narrow range of strain rates (e.g., $3 \times 10^{-1} – 3 \times 10^{-2}$ yr$^{-1}$ for a shear zone temperature of 240 K; Fig. 4b). By contrast, the variation in $n_{eff}$ derived for steady-state grain size from the wattmeter varies less dramatically with strain rate and is within the range of $3\pm0.5$ for all strain rates found in natural systems. Thus, the stress dependence of grain size evolution, when coupled to the composite flow law (Eq. 15), provides an explanation for why the effective stress exponent in ice is consistent with the Glen law, even though neither dislocation nor GBS-creep have stress exponents of ~3.

Further, when grain size evolves according to the wattmeter, smaller values of the grain growth exponent $p$ will result in larger values of $n_{eff}$ (Eq. 20), which becomes infinite when $p = (m_{GBS} – 1)$. This provides an additional argument against applying the small grain growth exponents for bubble-free ice in the laboratory to natural settings. For example, if $p = 2$ the effective stress exponent for GBS-limited creep becomes 4.25. In this scenario, neither dislocation creep nor GBS-limited creep would result in an effective stress exponent that is consistent with the Glen law value. This further validates our choice

of $p$ values consistent with the larger grain growth exponents inferred from bubble-rich experiments (Azuma et al., 2012). Intriguingly in the theoretical limit of grain growth in the presence of inclusions ($p = 3–4$; Evans et al., 2001) the effective stress exponent becomes 3.3–2.9 for steady-state grain size in the GBS regime.

Eq. (20) can also be used to predict the effective stress exponent for creep in other geologic materials that undergo grain-size sensitive creep and whose grain size evolution can be predicted by the wattmeter. For example, Hansen et al. (2012)

found that at a constant grain size GBS-creep in olivine is described by flow law parameters $n = 4.1$ and $m = 0.73$. However, in high strain experiments when grain size evolution occurred, the effective stress exponent increased to $n = 5$. Plugging the constant grain size parameters for GBS-creep into Eq (20) and assuming a grain-growth exponent of $p = 3$ for olivine (Karato, 1989), we calculate an effective stress exponent of $n_{eff} = 5.1$, consistent with the experimentally-determined value from the Hansen et al. (2012) experiments. This provides additional evidence that the wattmeter can be used to capture the physics of

grain-size sensitive creep.



## 4.2 Implications for grain size in ice cores

Ice cores show three primary grain size regimes (e.g., Gow & Williamson, 1976; Herron & Langway, 1982; Thorsteinson et al., 1997): (1) a zone of increasing grain size in the upper several hundred meters of ice, (2) a region of relatively constant to slightly decreasing grain size at intermediate depths, and (3) a zone of rapidly increasing grain size near the bed. These
variations have frequently been interpreted in terms of the *tripartite paradigm* or 3-stage model (e.g., Alley, 1988; 1992; De la Chapelle et al., 1998), in which Regime 1 is associated with normal grain growth, Regime 2 reflects a balance between normal grain growth and polygonization, and Regime 3 is attributed to migration recrystallization. The later process reflects a combination of rapid grain boundary migration and the nucleation of new grains when temperatures exceed 263 K (Duval & Castelnau, 1995).

More recent studies (e.g., Faria et al., 2014a) have argued that the tripartite model may be an oversimplification, as other processes besides normal grain growth appear to be operating at shallow depths (Kipfstudhl et al., 2006; 2009). Faria et al. (2014a) refer to the process by which grains coarsen while simultaneously undergoing deformation as "dynamic grain growth". The wattmeter inherently captures the balance between grain growth and grain size reduction, predicting grain sizes that vary continuously between Regime 1 and 2. However, as noted above, the wattmeter does not explain the increase in grain size
observed in Regime 3 near the base of the GISP2 core (Fig. 7) and other ice cores, such as Byrd (Gow & Williamson, 1976), GRIP (Thorsteinsson, et al., 1997), and Law dome (Jun et al., 1998). The reason is that the higher stresses and higher strain rates near the bed promote grain size reduction, which dominates the temperature-dependence of grain growth even as ice temperatures approach 263 K. One possible explanation for this discrepancy is that grain growth kinetics change as ice enters the pre-melting regime at temperatures > 263 K. For example, micro-particles on grain boundaries may become more mobile,
possible reducing their pinning effect and leading to enhanced grain growth (Evans et al., 2001).

As a simple test of this hypothesis, we substituted the bubble-free grain growth kinetics from Azuma et al. (2012) experiment T15 (conducted at 263 K) into the lowermost 200 m of our model for GISP2. The result is to increase grain sizes in the basal ice to ~100 mm. This is approximately an order of magnitude greater than the maximum observed values; however, using an intermediate grain growth exponent between ice with and without bubbles ($p$=4) provides a good fit to the observations
as shown by the dashed line in Fig. 7. While these results are suggestive, future work on grain growth kinetics in the pre-melting regime are needed to distinguish between these effects and the role of migration recrystallization in the formation of new grains (Duval & Castelnau, 1995; Hamann et al., 2007).

Another caveat of our predictions for grain size is that we have made no attempt to incorporate local scale heterogeneities in impurity contents. The role of impurities is well known to influence grain size in ice cores on multiple spatial/temporal
scales. At the centimeter-scale, "forest-fire" bands characterized by high ammonium contents and low electrical conductivities are observed to correlate with local reductions in grain size (e.g., Alley & Wood, 1996). Major climatic transitions, such as that associated with the Holocene/Last Glacial Maximum (LGM), are also seen to correlate with variations in grain size (e.g., Duval & Lorius, 1980; Herron et al., 1995; Gow et al., 1997; Jun et al., 1998) and zones of enhanced strain-rate (e.g., Fisher



& Koerner, 1986). Indeed, Durand et al. (2006) argue that grain growth pinned by a combination of dust, bubbles and clathrates
is the dominant control on grain size variability in Dome Concordia core. While incorporating heterogeneous impurity contents
is beyond the immediate scope of this study, the wattmeter provides a framework to include such heterogeneities through the
use of variable grain growth parameters tuned for different impurity contents. This further highlights the need for additional
grain growth experiments under various impurity contents and temperature conditions.

### 4.3 Grain size evolution and the origin of enhancement factors to the Glen Law

While the Glen law provides an excellent description of ice flow in many settings, certain systems are characterized by larger
strain rates than predicted. In such cases, an *ad hoc* strain enhancement factor is often incorporated into the pre-exponential
term of the Glen law to account for the combined effects of grain size, impurities, fabric development, and shear heating (c.f.,
Cuffey & Paterson, 2010). For example, matching velocity profiles across ice streams (e.g., Echelmeyer et al., 1994; Jackson
& Kamb, 1997) and through Pliestocene ice near the base of the Greenland ice sheet (Dahl-Jensen & Gunderstrup, 1987; Shoji
& Langway, 1988; Lüthi et al., 2002; Ryser et al., 2014) often requires enhancement factors in the range of 2–10. Cuffey et
al. (2000) attempted to quantify the role of grain size on the enhancement factor based on deformation recorded in Meserve
Glacier, Antarctica. The grain size evolution model developed here provides additional constraints on the role of grain size
on enhanced flow and strain localization in ice.

To illustrate this point, we model deformation within Drill Site D in fast moving ice near Jakobshavn Isbrae in western
Greenland (Iken et al., 1993; Lüthi et al., 2002). This site experiences surface velocities of ~600 m/yr and tiltmeter data
indicates enhanced strain rates in temperate ice below the Holocene–LGM transition near the bed. Lüthi et al. (2002) developed
a thermo-mechanical model for deformation in the borehole and found that after incorporating the temperature-dependence of
ice viscosity, enhancement factors of 1.7–2.6 were required to match the observations in the pre-Holocene ice below 680 m.
Although neither grain size nor impurity contents were measured in the Site D core, Lüthi et al. (2002) interpreted the enhanced
strain rates to reflect smaller grain sizes associated with higher impurity contents below the Holocene-LGM transition.

In Fig. 8 we apply the wattmeter to model deformation with the Site D using the same approach as for the GISP2 core
(Sect. 3.3) assuming a surface slope of 2º, an ice thickness of 830 m, downhole temperatures from Iken et al. (1993), and the
age model of Lüthi et al. (2002). Calculated grain sizes vary from ~2 mm near the surface to ~0.5 mm near the bed (Fig. 8a).
Comparing the corresponding strain-rates to those calculated for a case using a constant grain size of 1 mm, we predicted
enhancement factors of 1.9–2.5 in ice below ~700 m depth (Fig. 8c). Further, while there are no constraints on grain size for
direct comparison, the surface velocity calculated from our model compares favorably with those observed at the Site D
location (Fig. 8b). Thus, without invoking additional pinning effects beyond those incorporated in the grain growth exponents
extrapolated from the laboratory and GRIP ice core data (Fig. 2), the wattmeter provides a good match to the available
observations.

We stress that these results are not meant to imply that elevated impurity contents have no influence on grain size and
deformation rates, but simply that first-order variations in these parameters are successfully captured by the wattmeter.

Moreover, the enhanced strain rates associated with grain size reduction illustrate the potential importance of grain size evolution on strain localization. Indeed, the extreme strain localization in ice stream margins (e.g., Harrison et al., 1998) may be partially accommodated by grain size reduction, in combination with shear heating (e.g., Suckale et al., 2014; Perol & Rice,
2015). Future studies that simultaneously measure deformation, grain size, crystal fabric, and impurity contents—ideally in regions of high strain-rates—will be critical to improving coupled models of deformation and grain size evolution in ice sheets and glaciers.

## 5. Conclusions

We used the wattmeter (Austin & Evans, 2007; 2009) to calculate the balance between the mechanical work required for grain
growth and for dynamic grain size reduction. Combining the wattmeter with a composite flow law for dislocation and GBS creep, we developed a system of coupled equations that can be used to predict grain size evolution in terms of temperature, stress, and strain rate. Applying this methodology to grain sizes recorded in laboratory shear deformation experiments and the GISP2 borehole, we show that this approach successfully predicts grain size over a wide range of conditions.

When grain size evolution is accounted for using the wattmeter, we find that ice deforms with an effective stress exponent
of $n = 3.0 \pm 0.5$ at most natural conditions. This provides an explanation for the long-standing paradox of why the Glen law so successfully describes flow in glaciers and ice sheets, even though laboratory experiments show that neither dislocation creep nor GBS creep have stress exponents consistent with $n = 3$. Additionally, grain size variations driven by local deformation conditions can cause strain rate enhancement in regions where the Glen law alone cannot explain observed variations in ice flow. In conclusion, the coupling of grain size evolution and grain size sensitive creep, provides a potentially
powerful tool for understanding strain localization and the effective stress exponent in ice, as well as other geologic materials.

*Code availability.* MATLAB® code to reproduce Fig. 6 is provided as a supplement. All other code will be made available via GitHub after final acceptance of this manuscript.

*Author contributions.* All authors participated in the formulation of the grain size evolution model for ice. MB developed the model code and performed the simulations. DG and GH assisted in the interpretation of the model results. MB prepared the manuscript with contributions from all co-authors.

*Competing interests.* The authors declare that they have no competing interests.

*Acknowledgements.* We thank Andrew Cross for suggestions on an earlier version of this manuscript. Funding for this research was provided by NSF awards OPP-18-38410 and EAR-16-24109 (MB), and EAR-16-24178 (GH), and the NASA Solar System Workings program (NNX15AM69G) (DG).





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



**Table 1: Flow Law and Model Parameters**

| Symbol | Description | Value | Units |
|---|---|---|---|
| $n_{disl}$ | dislocation creep exponent | 4 | |
| $n_{GBS}$ | GBS creep exponent | 1.8 | |
| $A_{disl}$ | dislocation creep prefactor (>259 K, ≤259 K) | $6\times10^{28}$, $4\times10^{4}$ | MPa$^{-4}$ s$^{-1}$ |
| $A_{GBS}$ | GBS creep prefactor (>259 K, ≤259 K) | $3\times10^{26}$, $3.9\times10^{-3}$ | MPa$^{-1.8}$ s$^{-1}$ |
| $Q_{disl}$ | dislocation creep activation energy (>259 K, ≤259 K) | 180, 60 | kJ mol$^{-1}$ |
| $Q_{GBS}$ | GBS creep activation energy (>259 K, ≤259 K) | 192, 49 | kJ mol$^{-1}$ |
| $m_{disl}$ | dislocation creep grain-size exponent | 0 | |
| $m_{GBS}$ | GBS creep grain-size exponent | 1.4 | |
| $Q_{gg}$ | activation energy for grain growth | 42 | kJ mol$^{-1}$ |
| $K_{gg}$ | grain growth rate constant (lab, lab+ice core) | $2.58\times10^{-8}$, $1.02\times10^{-8}$ | m$^{p}$s$^{-1}$ |
| $p$ | grain growth exponent (lab, lab+ice core) | 7.1, 6.03 | |
| $\gamma$ | average specific grain boundary energy | 0.065 | J/m$^2$ |
| $\lambda_{disl}$, $\lambda_{GBS}$ | fraction of work done by dislocation and GBS creep to change grain boundary area | 0.005–0.05 | |
| $c$ | geometric constant | 3 | |


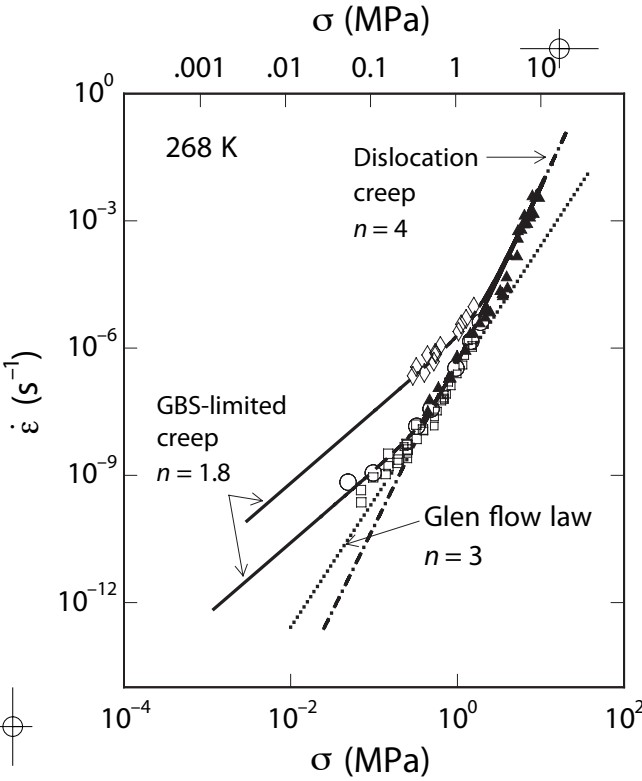


**Figure 1.** Strain-rate versus stress compiled from laboratory experiments on coarse-grained ice revealing the existence of the dislocation creep regime (n=4) and GBS-limited creep regime (n=1.8) at high and low stress, respectively. The upper and lower solid lines show grain boundary sliding flow law calculated for grain sizes of 0.2 and 2 mm, respectively; dashed-dot line shows dislocation creep flow law; dotted line depicts the Glen Law. Data are from ambient pressure tests at 268 K: d = 0.2 mm (diamonds) (Goldsby & Kohlstedt, 2001); d ≥ 1 mm (squares) (Steinemann, 1958); d ≥ 1 mm (circles) (Mellor & Smith, 1966); d ≥ 1 mm (triangles (Barnes et al. 1971). Note that the Glen law fails to adequately describe the flow of ice over a wide range of stresses. Figure adapted from Goldsby (2006).




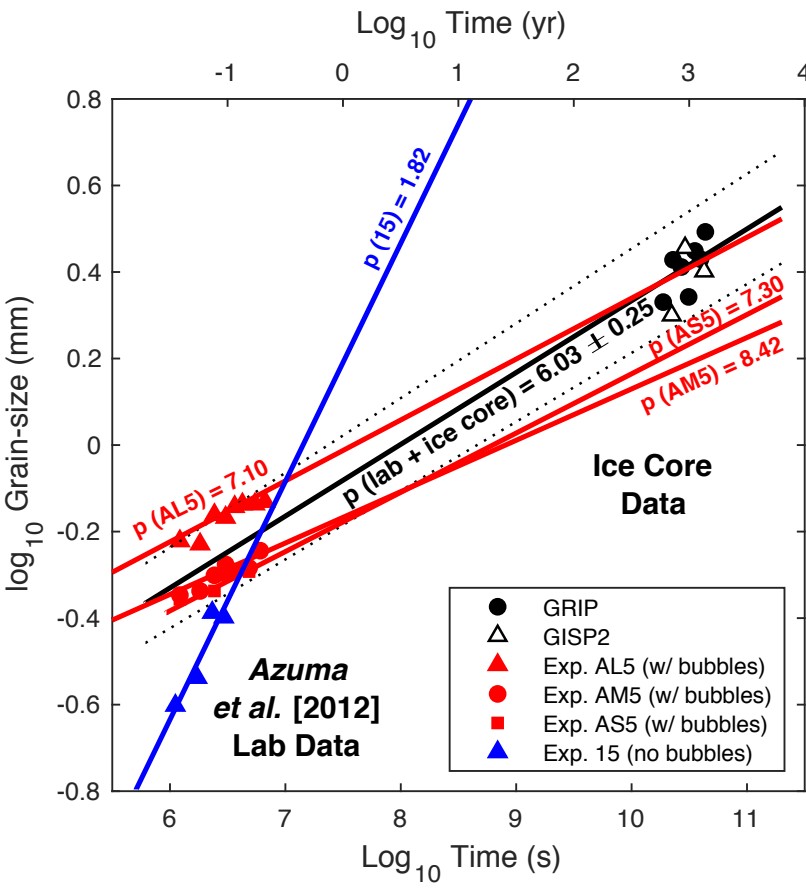

**Figure 2.** Comparison of grain growth rates derived from laboratory and ice core data. Data from individual laboratory
experiments by Azuma et al. (2012) with and without bubbles are shown by red and blue symbols, respectively. Grain sizes
for the GRIP (black circles; Thorsteinsson et al. (1997)) and GISP2 (open triangles; Gow et al. (1997)) ice cores are plotted as
a function of time based on the age models of Dansgaard et al. (1993) and Ram et al. (2000), respectively. Only ice core data
between 150–300 m depth where grain growth dominates is used (see text). Red and blue curves show fit to individual
experiments conducted at a temperature equivalent to the ice core data (243K); grain growth exponents (labeled) are calculated
following the methodology in Bons et al. (2001). Black curve shows fit calculated using all three laboratory experiments that
contain bubbles and the GRIP ice core data. Dotted black lines show 1-sigma error estimate on fit to lab and ice core data.






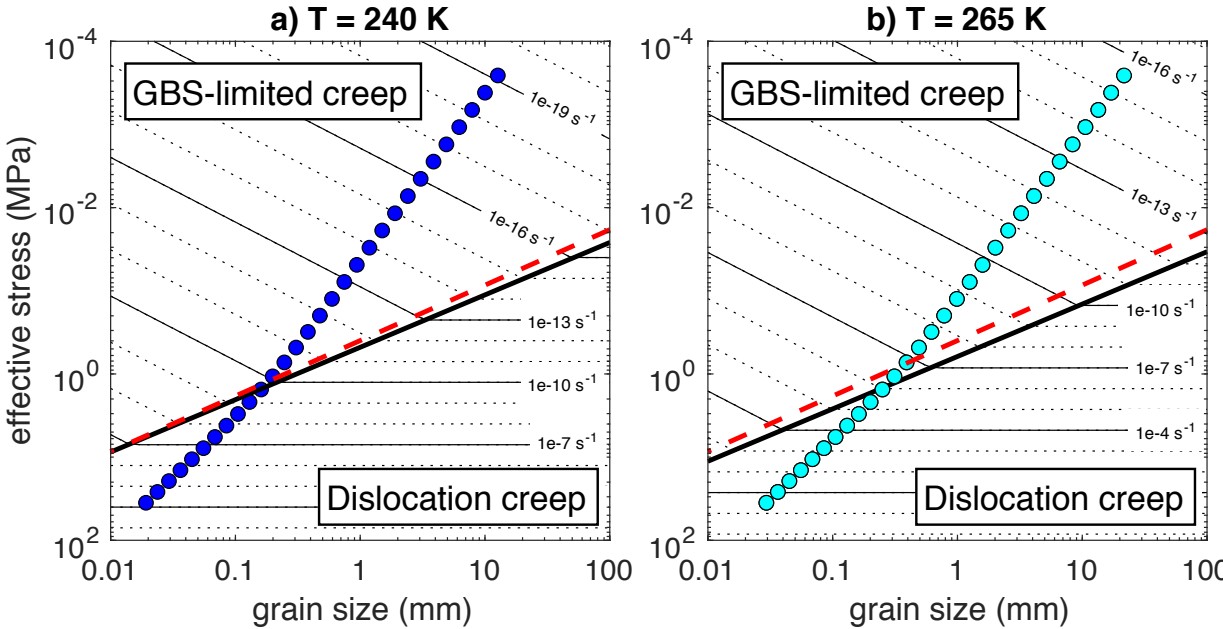

**Figure 3.** Effective stress vs. grain size at **(a)** 240K and **(b)** 265K calculated for a shear zone of fixed width using the
wattmeter. Dark and light blue symbols correspond to the steady-state grain size predicted from a single model simulation at
a given strain rate. Dashed red lines show location of the piezometer (Jacka & Jun, 1994). Model results are overlain on a
deformation mechanism map for ice calculated at the appropriate temperature using the flow law parameters from Goldsby &
Kohlstedt (2001). Background contours correspond to strain-rate; thick black line indicates the boundary between GBS-limited
creep (upper-left) and dislocation creep (lower-right). Under these conditions the location of the field boundary and piezometer
are very similar.


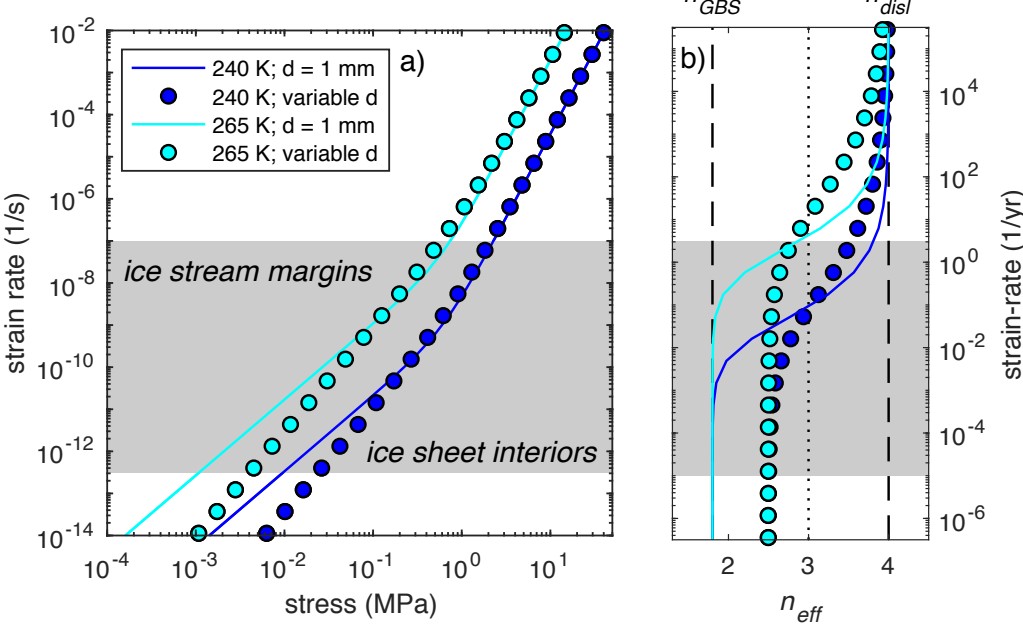

**Figure 4. (a)** Comparison of strain rate vs. stress predicted for a constant grain size of 1 mm (colored lines) to those predicted
by the steady-state grain size calculated from the wattmeter (colored symbols). Dark blue and light blue colors correspond to
temperatures of 240 K and 265 K, respectively. Wattmeter calculations correspond to those shown in Fig. 3 for a shear zone
of fixed width. **(b)** Effective stress exponent as a function of strain-rate predicted from the model. The effective stress exponent
is calculated from the slope of the strain-rate vs. stress curve shown in panel (a). For cases with a fixed grain size, the stress
exponent transitions from the experimentally-derived value for GBS-limited creep (at low strain rate) to the value for
dislocation creep (at high strain rate). The effective stress exponent in the GBS-limited creep regime calculated form the
wattmeter is higher than the experimentally-determined value and remains closer to the Glen law value of ~3 for strain rates
typical of natural systems.


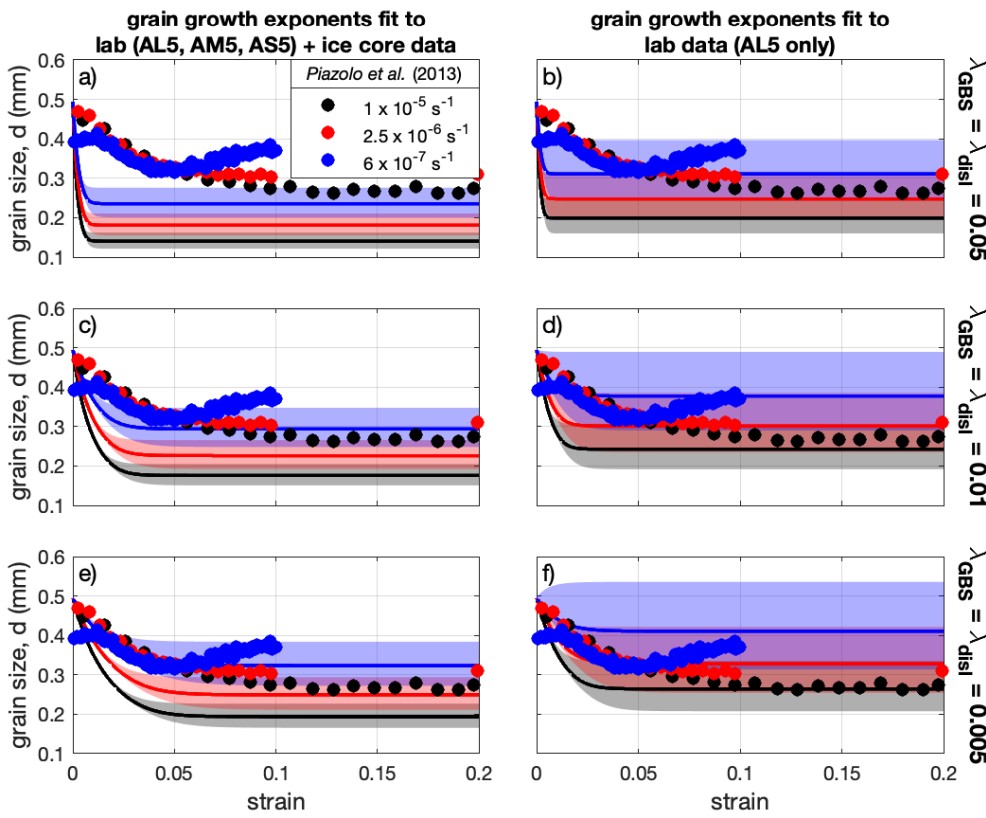

**Figure 5.** Comparison of wattmeter to experimental data on grain size evolution from Piazolo et al. (2013). Calculations are performed assuming a shear zone with an imposed strain rate corresponding to the laboratory experiments (1 x $10^{-5}$ s$^{-1}$ black; 2.5 x $10^{-6}$ s$^{-1}$ red; 6 x $10^{-7}$ s$^{-1}$ blue). Initial grain size is set to 0.5 mm and grain size evolution is calculated as a function of time/strain using Eq. (13). Panels **(a,c,e)** show results using the grain growth parameters from the joint fit between the laboratory and ice core data (black line in Fig. 2); panels **(b,d,f)** show results using the grain growth parameters from Azuma et al. (2012) experiment AL5. Rows indicate calculations using different values for $\lambda = \lambda_{GBS} = \lambda_{disl}$ ranging from **(a,b)** 0.05, **(c,d)** 0.01, and **(e,f)** 0.005.





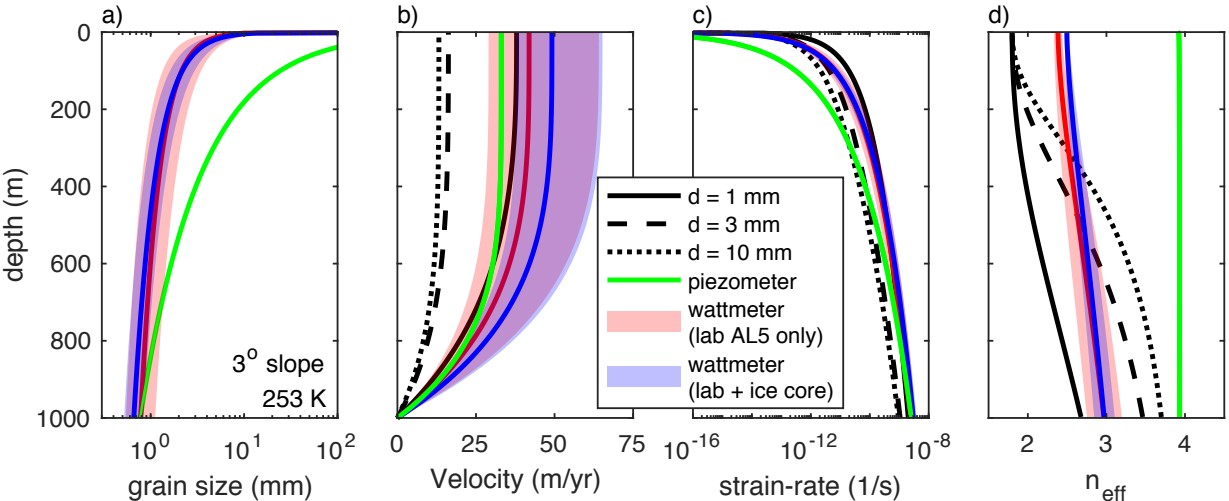

**Figure 6. (a)** Steady-state grain size, **(b)** velocity, **(c)** strain-rate, and **(d)** the effective stress exponent, $n_{eff}$ calculated as a function of depth. The effective stress exponent is calculated from our model using the local gradients in stress and strain rate. Red and blue curves correspond to calculations using the grain growth parameters from Azuma et al. (2012) experiment AL5 and the joint fit of the experimental and ice core data, respectively; shading denotes error bounds based on uncertainty in fit of the grain growth data. Black curves show calculates constant grain sizes of 1 mm (solid), 3 mm (dashed), and 10 mm (dotted). Green curve shows calculations based on the piezometer of Jacka & Jun (1994). Note that the effective stress exponents calculated using the wattmeter fall in a range similar to the Glen law ($n_{eff}$ = 2.5–3).







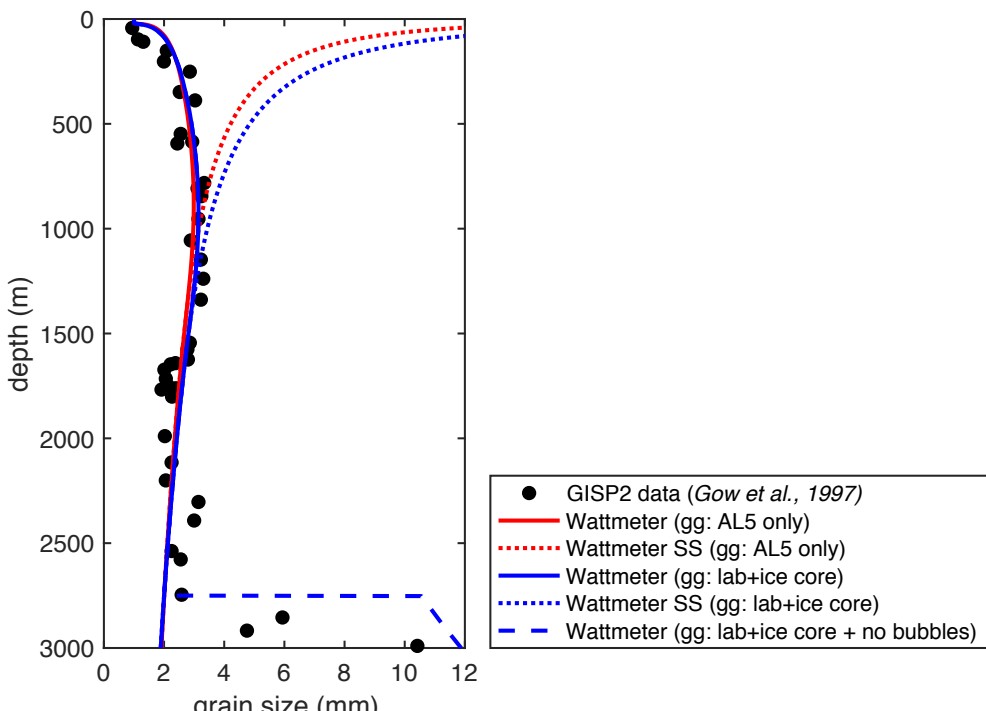

**Figure 7.** Grain size calculated as a function of depth within the GISP2 ice core. Red and blue curves correspond to
calculations using the grain growth parameters from Azuma et al. (2012) experiment AL5 and the joint fit of the experimental
and ice core data, respectively. Solid curves show time-dependent grain size calculations; dashed curves are the steady-state
grain size. Dashed curve shows calculation in which the modified bubble-free grain growth parameters (see text) are used in
the lowermost 200 m of ice. Black dots show observed grain sizes taken from Gow et al. (1997).


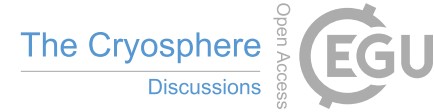


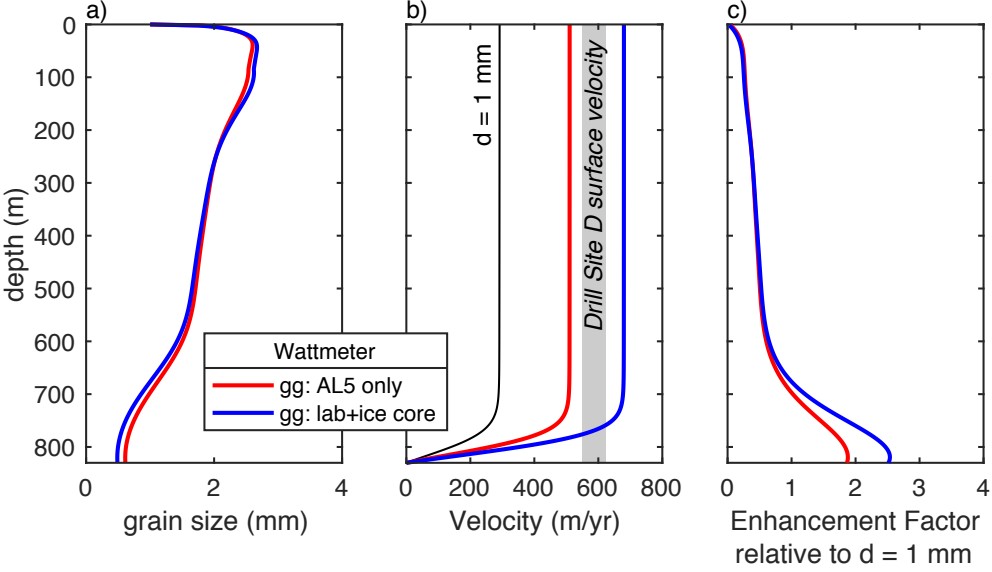


**Figure 8. (a)** Grain size, **(b)** velocity, and **(c)** strain-rate enhancement factor calculated as a function of depth for Drill Site D (Lüthi et al., 2002). Red and blue curves correspond to calculations using the grain growth parameters from Azuma et al.

(2012) experiment AL5 and the joint fit of the experimental and ice core data, respectively. Black curve in (b) corresponds to a case with a constant grain size of 1 mm. Enhancement factor is calculated as the ratio of the strain rate determined by the wattmeter to the strain rate calculated assuming a constant grain size of 1 mm.