# Peer review of "The role of grain-size evolution on the rheology of ice: Implications for reconciling laboratory creep data and the Glen flow law"

_The Cryosphere, 2020_

## Short Comment (SC1) · 22 Nov 2020

Behn et al. provide an interesting model to reconcile apparently contradictory laboratory creep data from ice samples and Glen's power law for glacier and ice-sheet flow. For the latter, a stress exponent of n=3 is usually assumed, as the authors correctly state. A stress exponent of n≈3 is actually also reported for many experiments (see for example Table 1 in Weertman, 1983). However, Goldsby and Kohlstedt (2001) and Goldsby (2006) presented experimental results that indicate that the stress exponent can achieve different values depending on the active deformation mechanism, with n=1.8 for grain boundary sliding and n=4 for dislocation creep. Glen (1955) actually

already noted that under natural conditions n should be higher than his experimentally determined n≈3.2. The main point of this comment is that a discrepancy between the stress exponent in nature and experiment may not exit. In a recent study, Bons et al. (2018) used data from a large area covering the northern part of the Greenland Ice Sheet and obtained a value of n≈4, which is consistent with dislocation glide according to experimental data by Goldsby and Kohlstedt (2001) and Goldsby (2006). There are only a few such studies that actually attempted to determine the stress exponent directly from ice sheets or glaciers. Instead, Glen's law with n=3 is usually assumed uncritically and other parameters, such as basal friction coefficients are derived instead. Behn et al. should therefore also consider the literature that suggests the possibility that n≈4 actually best describes the bulk rheology of ice sheets. The authors may also wish to consider the paper by Roessiger et al. (2014), which points out that the grain growth constant ($K_{gg}$ in Eq. 3) is not a constant, but depends on the microstructure. This has two relevant consequences for the paper under consideration. First, when the microstructure changes during an experiment, the resulting grain growth exponent (p in Eq. 2) is probably wrong if it is derived with the assumption that $K_{gg}$ is a constant. Second, it can be assumed that the microstructure is a function of the relative contribution of dislocation creep (n≈4) and grain boundary sliding (n≈1.8). If so, $K_{gg}$ will depend on that relative contribution. Taking the above into consideration would make the paper by Behn at al. much more interesting and worthwhile.

Paul Bons, Eberhard Karls University Tübingen, Germany

References

Bons, P. D., Kleiner, T., Llorens, M. G., Prior, D.J., Sachau, T., Weikusat, I. Jansen, D.: Greenland Ice Sheet – Higher non-linearity of ice flow significantly reduces estimated basal motion. Geophys. Res. Lett., 45, 6542-6548. doi:10.1029/2018GL078356, 2018.

Glen, J. W.: The creep of polycrystalline ice, Proceedings of the Royal Academy of

London Series A, 228(1175), 519–538, doi:10.1098/rspa.1955.0066, 1955.

Goldsby, D. L.: Superplastic Flow of Ice Relevant to Glacier and Ice-Sheet Mechanics, in Glacier Science and Environmental Change, edited by P. G. Knight, pp. 308–314, Blackwell Publishing, Malden, MA, USA., 2006.

Goldsby, D. L. and Kohlstedt, D.: Superplasticdeformation of ice: Experimental observations, J. Geophys. Res., 106(B6), 11,017–11,030, doi:10.1029/2000JB900336, 2001.

Roessiger, J., Bons, P. D., Faria, S. H.: Influence of bubbles on grain growth in ice, J. Struct. Geol., 61, 123-132. doi:10.1016/j.jsg.2012.11.00, 2014.

Weertman, J.: Creep deformation of ice, Ann. Rev. Earth Planet. Sci., doi:10.1146/annurev.ea.11.050183.001243, 1983.

---

## Referee Comment (RC1) · Anonymous Referee #1 · 31 Dec 2020

General comments The Behn et al. contribution tackles a persistent question in the glaciological community, namely whether the stress exponent of 3 typically used in ice flow constitutive laws has a robust physical justification, despite the lack of direct supporting experimental data (even in the classic Glen papers of the 1950s). The work is timely, of widespread interest, and with high potential impact. This study concludes that the combination of two deformation mechanisms – grain boundary sliding and dislocation creep – combine through a grain size dependency to produce a composite constitutive law with a stress exponent of approximately 3. I find the paper has a nice flow and clear macroscopic logic. At the same time, potentially ambiguous definitions and an incomplete treatment of uncertainty means that this paper as written is unlikely

to be perceived in the community as close to the final word on the topic. Restructuring the analysis to more explicitly incorporate uncertainty will take some time but yield a contribution with stronger impact.

Specific comments In this section, I list a number of focused comments and questions that I feel the authors should address in a revised version of this manuscript.

Although we all have a mental picture of grain size, for the quantitative approach taken in this manuscript, a more exact definition of the term would be useful. Is grain size the diameter of the equivalent area of a circle, or some other measure? Tied to that, the grain circularity is a potentially critical concept yet barely mentioned. The comparisons with natural data evaluated size (and even that not clearly with the same definitions), but not circularity. Granted, circularity may not have a large practical effect, but some evaluation would assuage concerns.

Stress is used throughout the manuscript, and it too would benefit from clearer definitions and identification of the relationships among the various forms applied. For example, piezometers/wattmeters typically use differential stress, but Equations 17-19 use shear stress and Figures 3 and 4 plot effective stress. In earlier equations, such as Eq 5, the form of stress isn't specified. In addition, I did not see a definition for "effective stress", though I presume it represents the square root of the second invariant of the deviatoric tensor. By defining these terms and explaining how the equations use the appropriate formulations (e.g., shear stress cannot directly be used in the flow law), confidence will be higher in the calculation results.

This manuscript implies strongly that the grain size is a better wattmeter than piezometer. That may be, but which approach better matches geologic/glaciologic reality remains unresolved in the community: despite the availability of the wattmeter for over a decade, many studies still rely on a piezometric approach. As such, a more explicit comparison in this paper seems warranted. The partial approach presented here has some merit, but feels incomplete and, in places, inaccurate. For example, the authors
present evidence for the wattmeter matching grain size data, and I agree in part. Taking Figure 5 for the moment, there is no illustrated scenario in which the model matches the experimental data for all three strain-rate cases. The highest strain rate (blue) never achieves a steady state, so we cannot really evaluate that case. The lowest strain rate case (black) and middle (red) are best fit with different lambda values, neither of which is what Austin and Evans use in their original study. Yet I don't see a treatment of this uncertainty in, for example, Fig 6. (At least, as far as I can understand from the text, the uncertainty shown in Fig 6 does not include a variation in lambda.) In addition, on line 122, the text implies that all internal energy goes to grain boundary area. That may be sufficiently accurate, but should be justified by exploring the potential of dislocation-driven energy variations after eq 19. The questions and notes I raise in this paragraph all lead to a concern that uncertainty in parameter values and applied processes preclude a robust conclusion about the stress exponent derived from the presented data.

Another significant discussion component that would lead towards a greater impact of this paper is a comparison of the effect of grain size against other rheological controls. The principle factor to address is anisotropy due to fabric development. The relationship will change with depth and affect the stress-strain-rate environment, and may affect grain size evolution.

Overall, my recommendations fall into two categories: (1) improving clarity of definitions and methods and (2) treating uncertainty more thoroughly. This topic is significant enough that a paper such as this could be one many people will rely on, once the readership can have more confidence in the conclusions.

Technical Comments Equations 17 and 18: I do not understand the source nor assigned value for viscosity; it doesn't appear in Table 1 and is somewhat at odds with the formulation of Eq. 16. I imagine I am missing something here, so an explanation would help.
Line 224: I do not follow how the iteration between Eqs 14 and 18 works in practice.

Figure 4: This is a relatively small comment, but can sow uncertainty. As I read it, the right panel is derived from the slope of the left panel. However, the right panel seems to have the same number of teal and blue dots. I am not clear how the authors calculate slope at the termini of the series of discrete points.

Lines 353-355: "In this scenario..." These two sentences feel to me to be a circular argument.

I concur with the comment provided by PD Bons that the manuscript should recognize that natural data do not necessarily require n=3, and in fact n=4 may be a more accurate representation.

TCD

---

## Referee Comment (RC2) · Paul D. Bons (Referee) · 9 Feb 2021

I already commented briefly in my unsolicited comment on the interesting and provocative manuscript of Behn et al. that proposes a novel (at least in glaciology, I believe) way to address the question of grain size in glaciers and ice sheets and its relationship with the rheology and stress exponent for power-law creep of ice. I was fortunate that by the time I was asked for a review, one thorough review was already published. I concur with the anonymous reviewer and need not repeat her/his comments.

I hope the manuscript by Behn et al. will be published in TC as it gives the community valuable food for thought. However, I would suggest to first address a few issues: 1)

[Figure]

Does the paradox on which the paper is based really exist? 2) Grain-growth parameters may be over-simplified. 3) The merits of alternative explanations for the grain size - stress relationship could be discussed more.

These issues are discussed in more detail below.

In the section starting at line 40, a crucial aspect is missing. Glen and some other authors made it very clear that their stress exponent was determined for the minimum strain rate/maximum stress and not for steady state. Comparing the low n ($\approx$3) at very low strain (about 1-3%!) with high-strain steady-state flow may be like comparing apples and oranges.

The manuscript is based on the "paradox" mentioned in line 59. Simply put it is postulated that experiments indicate a stress exponent n of either ca. 1.8 (low stress) or ca. 4 (high stress), while natural flow is closer to n=3, the value generally (and uncritically!) used in flow modelling. The question is whether this paradox really exists. In lines 28-34 it is argued that natural flow is consistent with n$\approx$3. Although several studies indeed come to this conclusion, others do not. For example Bons et al. (2018) deduced n$\approx$4 for a large area of the Greenland Ice Sheet (excluding the divides, ice-sheet margins and ice streams), while Pettit & Waddington (2003) find n$\approx$1 at divides. Glen (1955) himself wrote "... it is noteworthy that practically observable long-time creep rates, as in a glacier, would probably depend on a higher power of the stress than the 3.2 found here", although he did not actually determine this in natural ice. Cuffey and Kavanaugh (2011) write "we conclude that the effective n must be between 2.6 and 5.1 (99% confidence). The best match occurs with n $\approx$ 3.5". However, in the conclusions they also write "... supports the nearly universal practice of treating ice as an n = 3 nonlinear fluid in analyses of glacier flow". This may be symptomatic: despite evidence or indications to the contrary, some authors appear to (want to) stick to n=3, even if the data are inconclusive or allow alternatives. Another example is fig. 14 in Budd & Jacka (1989). They plot surface velocity/height against driving stress and find a best fit with a slope between n=3 and n=4. However, assuming n=3, they interpret the range in data in

terms of temperature differences. Close (re-) assessment of the literature shows that there is quite abundant evidence for n unequal to 3 for natural ice flow, even though the literature unfortunately does not always fairly acknowledge this. I suggest the authors: (1) qualify their basic starting assumption that natural ice follows n≈3 (2) and include in their following analysis what the consequences would be if n for natural flow is not 3, but perhaps indeed 4 as some claim to have measured in nature. Would this, for example, mean no contribution of GBS? Would the wattmeter work and give reasonable results?

In my unsolicited comment I already briefly addressed the grain-growth "constant" K and the grain-growth exponent p. The authors use p≈6, based on natural grain sizes in drill core and experiments with bubbly ice. There are a number of issues that I would ask the authors to consider.

(1) The exponent p reflects the scaling of the governing process(es). If grain growth is driven by unrestricted reduction of grain-boundary curvature and grain-boundary velocity is linearly proportional to the driving force (curvature), p should be 2. Restricted grain-boundary movement due to pinning or drag leads to a slow-down of growth, which gives a growth curve that may be fitted with a power law, but which is not a power law. The exponent p is "effective" or "apparent", but has little physical meaning and cannot be regarded as a universal material property. Growth then just does not follow a power law. If bubbles hinder growth, the effective p will depend on bubble size and distribution, relative to grain size (Arena et al., 1997; Roessiger et al. 2014). The main factor is probably the fraction of boundaries that is hindered in their movement by bubbles. If that fraction is small at the equilibrium grain size, the exponent p is expected to be close to 2, as most boundaries simply "don't know that they are in bubbly ice". In a grain-growth experiment that runs for long enough, one inevitably comes in the range where a significant number of boundaries interact with bubbles, which slows down the growth. The effective mobility of grain boundaries goes down, which raises the apparent p. This apparent p may not be relevant to the wattmeter if grain sizes are below this interaction

range. It should be noted that in the numerical simulations of Roessiger et al. (2014) p is always 2, just because of the scaling of the numerical simulations and governing equations. However, the growth curves would give a wide variety of p>2 values, if one would erroneously assume a power law.

(2) K is also not a universal constant, because it depends on the microstructure. This was actually one outcome of my very first paper: Bons & Urai (1992; I was so proud that I sent reprints to my whole family!). Static grain growth typically leads to a particular microstructure (grain shape and size distribution): a foam texture as in a soap froth. Changing the microstructure means changing K. Growth experiments are probably often hampered by this effect: it takes some growth to establish the steady-state growth rate. Measurements of K and p should only start after this is reached. Roessiger et al. (2014) therefore suggest a grain size increase of at least about 4-5 times. The resulting K is for static grain growth and does not apply to a dynamic grain-size equilibrium under consideration in the manuscript, where the microstructure is expected to be quite different. The distribution of bubbles may also be different during deformation compared to static experiments (Steinbach et al. 2016). It is not clear if a different, but constant K applies, or that K is a function of stress and/or strain rate.

The bottom line is that one should not consider a single, constant p and K. It is very well possible that p=2, but K varies depending on a variety of factors. How would this affect the analysis?

Line 83: " However, the piezometer does not account for the physical processes that control ice grain size - namely the competition between grain growth and grain-size reduction via recrystallization (e.g., Alley, 1992)." I suggest qualifying this rather sweeping sentence. There is a huge body of literature in materials science, metallurgy, geology, etc. on the physical processes that determine the piezometer. These models cannot be dismissed as "simple", nor do all say that grain size is the inverse of stress. The authors cite Jacka and Jun (1994). The authors of the paper are T.H. Jacka and Li Jun. The header of the original printed paper reads: "Jacka and Li: Steady-state crystal size of deforming ice". I therefore assume that the surname is "Li", not "Jun" and the Chinese convention of surname first was used. They do not find that grain size is inversely proportional to stress, but by an exponent of about -1.5. I do appreciate that the Jacka & Li piezometer is plotted in fig 3. It plots pretty much exactly on the boundary between the two deformation mechanisms as is acknowledged in the manuscript. So far, the data of Jacka & Li appear the only experimental grain size - stress data published in the literature and they would at first sight strongly support the de Bresser model. The slope of the piezometer is actually quite in line with that found for several other minerals, as pointed out by Jacka & Li and de Bresser et al (2001). Considering that natural flow of ice appears to be faster than experiments predict (compare the n=4 rates in Bons et al. (2018) with those used in the manuscript), the difference between gran size predicted by experiments and natural ice may be due to the infamous and "ad-hoc" enhancement factor. Line 284 is of interest: "Overall the piezometer [of Jacka & Li, 1994] results in smaller strain-rates throughout most of the column and a significantly higher effective stress exponent (neff $\sim$ 3.9), similar to the experimental value for dislocation creep." This n≈4 is exactly what is proposed by some authors for natural flow, which would fit very well with the piezometer. I suggest not to be too dismissive of the de Bresser model and the data of Jacka & Li (nor assume that natural flow has n=3).

Line 86: Typo in Roessiger Line 376: typo in Kipfstuhl

I hope these comments are not perceived as overly critical. The matter of the stress exponent is crucially important and far from trivial. All the more reason to be extremely careful. Only then may the community gain confidence in the rheological parameters it uses.

Paul Bons

References not listed in the reviewed manuscript, but that could be incorporated: Arena, L., Nasello, O.B., Levi, L., 1997. Effect of bubbles on grain growth in ice. Journal

of Physical Chemistry B 101, 6109-6112.

Bons, P.D., Urai J.L. 1992. Syndeformational grain growth: microstructures & kinetics. Journal of Structural Geology 14, 1101-1109. Doi:10.1016/0191-8141(92)90038-X.

Bons, P.D., Kleiner, T., Llorens, M.G., Prior, D.J., Sachau, T., Weikusat, I. Jansen, D. 2018. Greenland Ice Sheet . Higher non-linearity of ice flow significantly reduces estimated basal motion. Geophysical Research Letters 45, 6542-6548. Doi:10.1029/2018GL078356.

Budd, W. F., & Jacka, T. H. (1989). A review of ice rheology for ice sheet modeling. Cold Regions Science and Technology, 16, 107-144.

Cuffey, K. M., & Kavanaugh, J. L. (2011). How nonlinear is the creep deformation of polar ice? A new field assessment. Geology, 39, 1027-1030. Doi:10.1130/G32259.1.

Pettit, E. C., & Waddington, E. D. (2003). Ice flow at low deviatoric stress. Journal of Glaciology, 49, 359–369. Doi:10.3189/172756503781830584.

Roessiger, J., Bons, P. D., Faria, S. H.: Influence of bubbles on grain growth in ice, J. Struct. Geol., 61, 123-132. doi:10.1016/j.jsg.2012.11.00, 2014.

Steinbach, F., Bons, P.D., Griera, A., Jansen, D., Llorens, M.-G., Roessiger, J., Weikusat, I. 2016. Strain localisation and dynamic recrystallisation in the ice-air aggregate: A numerical study. Accepted for publication in The Cryosphere, 10, 3071-3089. doi:10.5194/tc-10-3071-2016

---

## Author Comment (AC1) · 15 Mar 2021

We thank Dr. Bons for his careful reading of our manuscript. Because he also provided a thoughtful formal review (in which he expands on these points). We address these issues in our response to his formal review below.

---

## Author Comment (AC2) · 15 Mar 2021

**Formal Review (Anonymous Referee #1, 31 Dec 2020):**

*General comments: The Behn et al. contribution tackles a persistent question in the glaciological community, namely whether the stress exponent of 3 typically used in ice flow constitutive laws has a robust physical justification, despite the lack of direct supporting experimental data (even in the classic Glen papers of the 1950s). The work is timely, of widespread interest, and with high potential impact. This study concludes that the combination of two deformation mechanisms – grain boundary sliding and dislocation creep – combine through a grain size dependency to produce a composite constitutive law with a stress exponent of approximately 3. I find the paper has a nice flow and clear macroscopic logic. At the same time, potentially ambiguous definitions and an incomplete treatment of uncertainty means that this paper as written is unlikely to be perceived in the community as close to the final word on the topic. Restructuring the analysis to more explicitly incorporate uncertainty will take some time but yield a contribution with stronger impact.*

Thanks for these suggestions. Below we describe specific changes that can be made to address the ambiguity in the definitions and the role of uncertainty in our model.

*Specific comments: In this section, I list a number of focused comments and questions that I feel the authors should address in a revised version of this manuscript.*
*Although we all have a mental picture of grain size, for the quantitative approach taken in this manuscript, a more exact definition of the term would be useful. Is grain size the diameter of the equivalent area of a circle, or some other measure? Tied to that, the grain circularity is a potentially critical concept yet barely mentioned. The comparisons with natural data evaluated size (and even that not clearly with the same definitions), but not circularity. Granted, circularity may not have a large practical effect, but some evaluation would assuage concerns.*

Grain size is a challenging material property to quantify because grains are irregular and are typically measured in a 2-D cross-section (e.g., thin section) through a 3-D sample. In practice grain size is measured in field and laboratory samples in a number of ways. Here we adapt the line intercept technique used by *Alley & Woods* (1996) in characterizing the GISP2 core. In this approach the average distance between grain boundaries along a series of lines through a sample is measured and then scaled by a correction factor of order 1 (=1.5 for circular grains; *Gifkens*, 1970) to account for the fact that when making a thin section many grains are cut near their edge as opposed to near their center (*Gow*, 1969). This approach was also used in the measurement of grain sizes in the derivation of the flow laws (*Goldsby & Kohlstedt*, 2001) and thus allows us to compare our calculated grain sizes to the GISP2 ice core data in a self-consistent manner. Thus, in both the wattmeter and our measurements we have implicitly assumed that grains are circular. This can be clarified in a revised manuscript.

Alley, R. B. and Woods, G. A., 1996, Impurity influence on normal grain growth in the GISP2 ice core, Greenland, J. Glaciol., 42(141), 255–260, doi:10.3189/S0022143000004111.
Gifkens, R.C., 1970, Optical Microscopy of Metals, American Elsevier, New York, 208 pp.
Goldsby, D. L. and Kohlstedt, D., 2001, Superplasticdeformation of ice: Experimental observations, J. Geophys. Res., 106(B6), 11,017–11,030, doi:10.1029/2000JB900336.
Gow, A.J., 1969. On the rate of growth of grains and crystals in south polar firn. Journal of Glaciology 8 (53), 241-252.

*Stress is used throughout the manuscript, and it too would benefit from clearer definitions and identification of the relationships among the various forms applied. For example, piezometers/wattmeters typically use differential stress, but Equations 17-19 use shear stress and Figures 3 and 4 plot effective stress. In earlier equations, such as Eq 5, the form of stress isn't specified. In addition, I did not see a definition for "effective stress", though I presume it represents the square root of the second invariant of the deviatoric tensor. By defining these terms and explaining how the equations use the appropriate formulations (e.g., shear stress cannot directly be used in the flow law), confidence will be higher in the calculation results.*

The reviewer is correct that we were somewhat sloppy in our definition of stress in the manuscript. The stress used in the definition of the wattmeter (defined as sigma in Eqs 5–14) and in the flow law (Eq. 16) is the Von Mises equivalent stress. However, in our simplified geometries the only non-zero term of the deviatoric stress tensor is the shear stress (defined as tau in Eqs 17–19). We relate the shear stress to the Von Mises stress through the square root of the second invariant of the stress tensor, which in this geometry simply reduces to $\tau = \sqrt{3}\sigma$.

*This manuscript implies strongly that the grain size is a better wattmeter than piezometer. That may be, but which approach better matches geologic/glaciologic reality re- mains unresolved in the community: despite the availability of the wattmeter for over a decade, many studies still rely on a piezometric approach. As such, a more explicit comparison in this paper seems warranted. The partial approach presented here has some merit, but feels incomplete and, in places, inaccurate. For example, the authors present evidence for the wattmeter matching grain size data, and I agree in part. Taking Figure 5 for the moment, there is no illustrated scenario in which the model matches the experimental data for all three strain-rate cases. The highest strain rate (blue) never achieves a steady state, so we cannot really evaluate that case. The lowest strain rate case (black) and middle (red) are best fit with different lambda values, neither of which is what Austin and Evans use in their original study. Yet I don't see a treatment of this uncertainty in, for example, Fig 6. (At least, as far as I can understand from the text, the uncertainty shown in Fig 6 does not include a variation in lambda.)*

To better illustrate the influence of the assumed lambda value (the fraction of the total work responsible for increases in internal energy), we have added a new row of panels to Figure 6. These panels illustrate the changes in grain size, velocity, strain-rate, and effective stress exponent that result from varying lambda over the range of values used in the comparison to the laboratory data (Figure 5). We emphasize that the differences in the wattmeter predictions due to the uncertainty in lambda are significantly smaller than the variations associated with the uncertainty in the grain growth parameters. The revised version of Figure 6 (below) will be included in the revised manuscript. As pointed out by *Austin & Evans* (2007; 2009), lambda values from ~0 to 0.4 are reported and these are not well constrained; they emphasize that lambda is a scaling factor. Similar values to what we report for ice have been determined by applying the wattmeter to recrystallization of quartzite (*Tokle et al.*, in revision) and olivine (*Holtzman* et al., 2018).

Austin, N. and Evans, B., 2009, The kinetics of microstructural evolution during deformation of calcite, J. Geophys. Res., 114(B9), B09402, doi:10.1029/2008JB006138.

Austin, N. J. and Evans, B., 2007, Paleowattmeters: A scaling relation for dynamically recrystallized grain size, Geol., 35(4), 343, doi:10.1130/G23244A.1.

Holtzman, B.K., Chrysochoos, A., and Daridon, L., 2018. A thermodynamical framework for analysis of microstructural evolution: Application to olivine rocks at high temperature. Journal of Geophysical Research: Solid Earth 123, 8474-8507. doi:10.1029/2018JB015613.

Tokle, L. and Hirth, G., 2020. Assessment of quartz grain growth and the application of the wattmeter to predict recrystallized grain sizes (submitted to JGR, available on EarthArXiv: https://eartharxiv.org/repository/view/1865/)

*In addition, on line 122, the text implies that all internal energy goes to grain boundary area. That may be sufficiently accurate, but should be justified by exploring the potential of dislocations and elasticity to store energy. Austin and Evans (2009), for example, mention dislocation-driven energy variations after eq 19. The questions and notes I raise in this paragraph all lead to a concern that uncertainty in parameter values and applied processes preclude a robust conclusion about the stress exponent derived from the presented data.*

[Figure]

By setting $\lambda_{GBS} = \lambda_{disl}$ in all our models, the effects of grain boundary energy, grain geometry, and lambda, can all be grouped into a single "scaling factor" $= \frac{c\gamma}{\lambda}$ in Eq. (14). Thus, the variations in lambda shown in the new version of Figure 6 elucidate the behavior of the model with respect to variations in any of these parameters. We also note that while we use a lower value of lambda compared to *Austin & Evans* (2007; 2009), the ability of the wattmeter to fit grain size data across multiple orders of magnitude in stress and strain-rate from the lab (Fig. 5)

to the GISP2 ice core data (Fig. 7) provides strong motivation that this approach is capturing the first-order physics of grain size evolution.

It is important to emphasize that the wattmeter formulation models the rate of *change* in the internal energy (with the lambda scaling factor), and relates this to the grain size reduction rate (and thus increase in internal energy owing to increase in grain boundary area). This is a bit confusing, because the driving force for the grain size reduction is indeed the internal energy associated with dislocations. The key assumption here, outlined by *Austin and Evans* (2007; 2009), is that the rate of change in grain size is greater than the rate of change in stress – thus the dislocation density can be considered constant for a given stress.

*Another significant discussion component that would lead towards a greater impact of this paper is a comparison of the effect of grain size against other rheological controls. The principle factor to address is anisotropy due to fabric development. The relationship will change with depth and affect the stress-strain-rate environment, and may affect grain size evolution.*

This is an excellent point. The development of fabric in the shear plane will weaken the ice, resulting in a larger strain rate for the same stress. Because the stress is fixed by the surface slope, this will result in a positive feedback in which enhanced fabric development will drive further grain size reduction (due to the enhanced work rate). Future studies should investigate these feedbacks in order to quantify the magnitude of this effect.

*Technical Comments Equations 17 and 18: I do not understand the source nor assigned value for viscosity; it doesn't appear in Table 1 and is somewhat at odds with the formulation of Eq. 16. I imagine I am missing something here, so an explanation would help.*

The viscosity is simply reformulated from the flow law (Eq. 16) in terms of the stress, e.g.,

$$\eta(\sigma) = A^{-1}\sigma^{1-n}d^m \exp\left(\frac{Q}{RT}\right)$$

*Line 224: I do not follow how the iteration between Eqs 14 and 18 works in practice.*

To calculate grain size, we first make a guess at the initial shear stress and grain size. Using these values, we calculate viscosity and use Eq (18) to make a new estimate of the shear stress. Based on our new estimate of shear stress and the corresponding strain-rate (calculated from the flow law), we use the wattmeter to calculate an updated steady-state grain size (Eq. 14). These new estimates for stress and grain size are then used to recalculate viscosity, which is in turn fed back in Eq (18) for the next iteration. We continue to iterate in this manner until the shear stress varies by less than 0.1%.

*Figure 4: This is a relatively small comment, but can sow uncertainty. As I read it, the right panel is derived from the slope of the left panel. However, the right panel seems to have the same number of teal and blue dots. I am not clear how the authors calculate slope at the termini of the series of discrete points.*

We actually made calculations at a much finer resolution in stress than is shown on the figure, thus while the dots are indeed slightly offset (by half the spacing in our sampling of stress) it is hard to see this in the figure. For clarity, we will adjust the figure to plot the dots in panel (b) halfway between the dots in panel (a).

*Lines 353-355: "In this scenario..." These two sentences feel to me to be a circular argument. I concur with the comment provided by PD Bons that the manuscript should recognize that natural data do not necessarily require n=3, and in fact n=4 may be a more accurate representation.*

Please see our response to Dr. Bons below, in which we address these concerns.

---

## Author Comment (AC3) · 15 Mar 2021

**Formal Review (Paul Bons, 9 Feb 2021):**

*I already commented briefly in my unsolicited comment on the interesting and provocative manuscript of Behn et al. that proposes a novel (at least in glaciology, I believe) way to address the question of grain size in glaciers and ice sheets and its relationship with the rheology and stress exponent for power-law creep of ice. I was fortunate that by the time I was asked for a review, one thorough review was already published. I concur with the anonymous reviewer and need not repeat her/his comments.*

*I hope the manuscript by Behn et al. will be published in TC as it gives the community valuable food for thought. However, I would suggest to first address a few issues: 1) Does the paradox on which the paper is based really exist? 2) Grain-growth parameters may be over-simplified. 3) The merits of alternative explanations for the grain size - stress relationship could be discussed more.*

*These issues are discussed in more detail below.*

Thank you for your thorough reading of our manuscript. Below we respond in detail to the issues that you have raised.

*In the section starting at line 40, a crucial aspect is missing. Glen and some other authors made it very clear that their stress exponent was determined for the minimum strain rate/maximum stress and not for steady state. Comparing the low n (≈3) at very low strain (about 1-3%!) with high-strain steady-state flow may be like comparing apples and oranges.*

*The manuscript is based on the "paradox" mentioned in line 59. Simply put it is postulated that experiments indicate a stress exponent n of either ca. 1.8 (low stress) or ca. 4 (high stress), while natural flow is closer to n=3, the value generally (and uncritically!) used in flow modelling. The question is whether this paradox really exists. In lines 28- 34 it is argued that natural flow is consistent with n≈3. Although several studies indeed come to this conclusion, others do not. For example, Bons et al. (2018) deduced n≈4 for a large area of the Greenland Ice Sheet (excluding the divides, ice-sheet margins and ice streams), while Pettit & Waddington (2003) find n≈1 at divides. Glen (1955) himself wrote "... it is noteworthy that practically observable long-time creep rates, as in a glacier, would probably depend on a higher power of the stress than the 3.2 found here", although he did not actually determine this in natural ice. Cuffey and Kavanaugh (2011) write "we conclude that the effective n must be between 2.6 and 5.1 (99% confidence). The best match occurs with n ≈ 3.5". However, in the conclusions they also write "... supports the nearly universal practice of treating ice as an n = 3 nonlinear fluid in analyses of glacier flow". This may be symptomatic: despite evidence or indications to the contrary, some authors appear to (want to) stick to n=3, even if the data are inconclusive or allow alternatives. Another example is fig. 14 in Budd & Jacka (1989). They plot surface velocity/height against driving stress and find a best fit with a slope between n=3 and n=4. However, assuming n=3, they interpret the range in data in terms of temperature differences. Close (re-) assessment of the literature shows that there is quite abundant evidence for n unequal to 3 for natural ice flow, even though the literature unfortunately does not always fairly acknowledge this. I suggest the authors: (1) qualify their basic starting assumption that natural ice follows n≈3 (2) and include*

*in their following analysis what the consequences would be if n for natural flow is not 3, but perhaps indeed 4 as some claim to have measured in nature. Would this, for example, mean no contribution of GBS? Would the wattmeter work and give reasonable results?*

This is a fair criticism. As Dr. Bons correctly points out, observational studies find a range of values for the effective stress exponent in deforming ice sheets and glaciers. Our model provides a framework in which to interpret such variations depending on the relative contributions of dislocation creep and grain boundary sliding—which are in turn modulated by the grain size. This point is illustrated in Figure 4b, where we show how the effective stress exponent can vary as a function of strain rate. The goal of our study is really to emphasize the point expressed in Dr. Bon's initial comment (22-Nov-2020) that "*Glen's law with n=3 is usually assumed uncritically and other parameters, such as basal friction coefficients are derived instead.*" This is an important point, because if variations in grain size and/or other aspects of the flow regime result in an effective stress exponent that deviates from the Glen law value, then any other parameters (e.g., basal friction) derived from models that assume n=3 will be incorrect.

*In my unsolicited comment I already briefly addressed the grain-growth "constant" K and the grain-growth exponent p. The authors use p≈6, based on natural grain sizes in drill core and experiments with bubbly ice. There are a number of issues that I would ask the authors to consider.*
*(1) The exponent p reflects the scaling of the governing process(es). If grain growth is driven by unrestricted reduction of grain-boundary curvature and grain-boundary velocity is linearly proportional to the driving force (curvature), p should be 2. Restricted grain-boundary movement due to pinning or drag leads to a slow-down of growth, which gives a growth curve that may be fitted with a power law, but which is not a power law. The exponent p is "effective" or "apparent", but has little physical meaning and cannot be regarded as a universal material property. Growth then just does not follow a power law. If bubbles hinder growth, the effective p will depend on bubble size and distribution, relative to grain size (Arena et al., 1997; Roessiger et al. 2014). The main factor is probably the fraction of boundaries that is hindered in their movement by bubbles. If that fraction is small at the equilibrium grain size, the exponent p is expected to be close to 2, as most boundaries simply "don't know that they are in bubbly ice". In a grain- growth experiment that runs for long enough, one inevitably comes in the range where a significant number of boundaries interact with bubbles, which slows down the growth. The effective mobility of grain boundaries goes down, which raises the apparent p. This apparent p may not be relevant to the wattmeter if grain sizes are below this interaction range. It should be noted that in the numerical simulations of Roessiger et al. (2014) p is always 2, just because of the scaling of the numerical simulations and governing equations. However, the growth curves would give a wide variety of p>2 values, if one would erroneously assume a power law.*

We agree with the argument here—namely, that the power law exponent reflects an "apparent" exponent, or perhaps is better thought of as an empirical exponent. The key here is that a power law is not necessarily "erroneous" – as it is an empirical fit. Where problems can arise is when the empirical relationship breaks down when extrapolated outside of the range of conditions where it was quantified. Determining how to deal with this was actually one of the most difficult aspects of this study. In the end, we were struck by the correspondence of how well the extrapolation of the power-law fit predicted the grain size in the shallow parts of the ice sheets

(where dynamic recrystallization is not active) – which supports applying the empirical p value, but indeed does not prove its applicability.

As pointed out by Dr. Bons and *Arena et al.* (1997) the role of pores can also be thought of as changing the *K* value in the grain growth law. If *K* varies with the microstructure (bubble size / bubble topology) and this scales with grain size, then *K* will be proportional to some function of grain size *f(d)*. In our formulation, the empirically fit p-value is mathematically similar to a *K* term with a power-law relationship to grain size.

Thus, while we acknowledge the importance of more sophisticated models that specifically account for how bubble mobility and bubble size impact grain growth, with the caveats described above, we note that an effective grain growth exponent of order 6–7 can fit both the laboratory and ice core data (Figure 2), as well as the grain growth in the shallow part of the GISP2 ice core (Figure 7).

A revised manuscript will provide a more detailed description of the assumptions that have gone into our parameterization of the empirical grain growth exponent, as well as a discussion of potential future theoretical developments in this area.

Arena, L., Nasello, O. B. & Levi, L., 1997, Effect of Bubbles on Grain Growth in Ice. *J Phys Chem B*, v. 101, 6109–6112.

*(2) K is also not a universal constant, because it depends on the microstructure. This was actually one outcome of my very first paper: Bons & Urai (1992; I was so proud that I sent reprints to my whole family!). Static grain growth typically leads to a particular microstructure (grain shape and size distribution): a foam texture as in a soap froth. Changing the microstructure means changing K. Growth experiments are probably often hampered by this effect: it takes some growth to establish the steady-state growth rate. Measurements of K and p should only start after this is reached. Roessiger et al. (2014) therefore suggest a grain size increase of at least about 4-5 times. The resulting K is for static grain growth and does not apply to a dynamic grain-size equilibrium under consideration in the manuscript, where the microstructure is expected to be quite different. The distribution of bubbles may also be different during deformation compared to static experiments (Steinbach et al. 2016). It is not clear if a different, but constant K applies, or that K is a function of stress and/or strain rate.*
*The bottom line is that one should not consider a single, constant p and K. It is very well possible that p=2, but K varies depending on a variety of factors. How would this affect the analysis?*

Following on the points made above, we can describe these issues as part of the assessment of uncertainty in how our assumption of the power law form of the grain growth law impacts our interpretations. As a starting point, it is instructive to compare our model predictions with data for grain growth under bubble-free conditions. In doing so, we are essentially assuming that the "drag-drop" condition leads to no hinderance of grain boundary mobility. Extrapolation of the bubble-free grain growth data to natural conditions predicts extremely large grain sizes. For example, using the grain growth parameters from *Azuma et al.* (2012) Exp. 15 our application of the wattmeter predicts grain sizes > 100 mm at all depths in the GISP2 core. These predictions

are of course significantly larger than the observed grain sizes, emphasizing that natural settings likely still include limited mobility owing to pinning, which is captured in our use of an empirical grain growth exponent.

Azuma, N., Miyakoshi, T., Yokoyama, S. and Takata, M., 2012, Impeding effect of air bubbles on normal grain growth of ice, J. Struc. Geol., 42(C), 184–193, doi:10.1016/j.jsg.2012.05.005.

*Line 83: " However, the piezometer does not account for the physical processes that control ice grain size - namely the competition between grain growth and grain-size re- duction via recrystallization (e.g., Alley, 1992)." I suggest qualifying this rather sweeping sentence. There is a huge body of literature in materials science, metallurgy, geology, etc. on the physical processes that determine the piezometer. These models cannot be dismissed as "simple", nor do all say that grain size is the inverse of stress. The authors cite Jacka and Jun (1994). The authors of the paper are T.H. Jacka and Li Jun. The header of the original printed paper reads: "Jacka and Li: Steady-state crystal size of deforming ice". I therefore assume that the surname is "Li", not "Jun" and the Chinese convention of surname first was used. They do not find that grain size is inversely proportional to stress, but by an exponent of about -1.5. I do appreciate that the Jacka & Li piezometer is plotted in fig 3. It plots pretty much exactly on the boundary between the two deformation mechanisms as is acknowledged in the manuscript. So far, the data of Jacka & Li appear the only experimental grain size - stress data published in the literature and they would at first sight strongly support the de Bresser model. The slope of the piezometer is actually quite in line with that found for several other minerals, as pointed out by Jacka & Li and de Bresser et al (2001). Considering that natural flow of ice appears to be faster than experiments predict (compare the n=4 rates in Bons et al. (2018) with those used in the manuscript), the difference between gran size predicted by experiments and natural ice may be due to the infamous and "ad-hoc" enhancement factor. Line 284 is of interest: "Overall the piezometer [of Jacka & Li, 1994] results in smaller strain-rates throughout most of the column and a significantly higher effective stress exponent (neff ~ 3.9), similar to the experimental value for dislocation creep." This n≈4 is exactly what is proposed by some authors for natural flow, which would fit very well with the piezometer. I suggest not to be too dismissive of the de Bresser model and the data of Jacka & Li (nor assume that natural flow has n=3).*

Both of the points are fair, and can be addressed with a more thorough discussion of the limitations and assumptions involved with the application of piezometers. The advantage of the wattmeter is that it provides scaling relationships for both steady state and transient grain sizes. The basic components of the wattmeter (as a competition between grain growth and grain size reduction) are actually based on the same logic as the piezometer model of *Jacka & Li* (1994), with the added insight that the grain size reduction processes involve the increase in internal energy within the crystals that drive grain size reduction – and how these depend on the product of stress times strain rate.

Regarding the field boundary hypothesis of de *Bresser et al*. (2001) – the basic idea to that model is that the driving force for grain boundary reduction becomes negligible when diffusion creep dominates. However, this is not applicable for the field boundary between DisGBS and dislocation creep, where easy slip on the basal plane of ice will produce intracrystalline deformation, similar to observations in olivine (e.g., *Hansen et al*., 2012). These observations

suggest that the similarity with the de Bresser field boundary may simply be a "compelling" coincidence – but certainly one worth noting in the revised the manuscript.

de Bresser, J., Ter Heege, J. and Spiers, C.: Grain size reduction by dynamic recrystallization: can it result in major rheological weakening?, Int. J. Earth Sci., 90, 28–45, doi:10.1007/s005310000149, 2001.
Hansen, L. N., Zimmerman, M. E. and Kohlstedt, D. L.: The influence of microstructure on deformation of olivine in the grain-boundary sliding regime, J. Geophys. Res., 117(B9), doi:10.1029/2012JB009305, 2012.
Jacka, T. H. and Li, J.: The steady-state crystal size of deforming ice, Ann. Glaciol., 20, 13–18, doi:10.3189/1994AoG20-1-13-18, 1994.

*Line 86: Typo in Roessiger Line 376: typo in Kipfstuhl*

Thanks – will be corrected

---

## Referee Report (RR1)

I thank the authors for the extensive replies to the comments of both reviewers. The authors did acknowledge that there were issues that needed to be addressed and made changes to the text accordingly. This definitively did improve the manuscript. However, changes in the manuscript are at times rather brief and do not quite reflect the more extensive discussion in the comments & replies. I will only focus on two issues: the actual value of the stress exponent and the effect of a crystallographic preferred orientation (CPO).

The authors now do acknowledge (lines 35-44 in document with show changes) that although $n$=3 is normally used, actual observations do not always agree with it, referring to Cuffey & Kavanaugh (2011) and Budd & Jacka (1989). There are, of course, more papers (not cited) that suggest that n is unequal to 3, like $n$=4. The origin of $n$=3 is not really acknowledged: taking the minimum strain rate or maximum stress point, which is at only a few per cent of strain. This paper is about higher strains. Having added the few sentences, the manuscript continues as it was, i.e. effectively based on the assumption that $n$=3 is not only commonly used, but indeed correct. The suggestion from the original review to *"include in their following analysis what the consequences would be if n for natural flow is not 3, but perhaps indeed 4 as some claim to have measured in nature. Would this, for example, mean no contribution of GBS? Would the wattmeter work and give reasonable results?"* is not considered.

On the contrary: In line 432, the authors write: *" This provides an additional argument against applying the small grain growth exponents for bubble-free ice in the laboratory to natural settings. For example, if p = 2 the effective stress exponent for GBS-limited creep becomes 4.25. In this scenario, neither dislocation creep nor GBS-limited creep would result in an effective stress exponent that is consistent with the Glen law value."* The authors argue thus that $p$ should not be small, because the results would then not fit with $n$=3 of Glen's law, which is apparently taken as correct. If one acknowledges that $n$ could be 4 (as I would say the actual velocity field of the Greenland ice sheet indicates), the conclusion would clearly be that $p$ should be small, because it nicely fits the observations. This again shows that the growth law is a big uncertainty of the model.

A sceptical or malicious reader could interpret this as a circular argument: first fit the parameters to get $n$=3 and then claim success of the model, because it fits $n$=3. One could also come to the conclusion that if the model works for $n$=3 while ignoring mechanical anisotropy (see below) it must be wrong, since $n$=3 only applies to circa 3% strain and because ice is anisotropic. To avoid such unwanted interpretations, it would be good (or even imperative) if the authors would consider how well the wattmeter would work if $n$ would be different, for example about 4.

The model focuses strongly (completely) on grain size as modifier of the effective viscosity. A big issue, raised in the review process, is actually the mechanical anisotropy of ice, which can greatly change the effective viscosity. After the review, the authors added only 2 short sentences (lines 522-524) addressing this elephant in the room. This I find rather meagre. What would be the effect of further grain size reduction that is mentioned? A lowering of the effective $n$ for

the whole ice sheet, if at the base where most shearing happens we may expect strong CPOs? Does that fit with some observations that grain size increases near the base? Or do other parameters need to be readjusted to refit the model to observations?

The wattmeter approach is of interest, as has already been proven for rocks other than ice, and therefore the paper could make a valuable contribution to glaciology. Without truly addressing alternatives to the isotropic Glen's law model, I am afraid it may be dismissed by those that acknowledge that that model is not realistic.

I therefore suggest publication after addressing these comments.

Kind regards, Paul Bons

---

## Author Response (AR2)

***Reviewer #1 (Anonymous)***

*I appreciate the authors responding to the comments and addressing the concerns raised. The manuscript is distinctly clearer in those areas. I remain ambivalent about whether the proposed approach solves the issue of the stress exponent in ice, given the uncertainty in the parameters inherent in the analysis (e.g., grain growth), the assumptions around grain shape and its measurement, and the lack of community consensus about grain size being a better piezometer or wattmeter. Despite that ambivalence, this paper adds to the discussion, and I think the community will benefit from its perspective.*

*I have only small recommended changes (line #s reference the track changes document):*

*Line 36: "report" to "reported"*

*Line 522: Duvall to Duval*

*Fig 6 caption (line 847) "Black curves show calculates constant..." to "Black curves show constant..."*

*Fig 6 caption (line 849): "(e-h) Same as panels a-d, but comparing..." to "(e-h) Same panel axes as a-d, comparing..."*

We thank Reviewer #1 for their careful reading of our revised manuscript. All minor points have been corrected.

***Reviewer #2 (Paul Bons)***

*I thank the authors for the extensive replies to the comments of both reviewers. The authors did acknowledge that there were issues that needed to be addressed and made changes to the text accordingly. This definitively did improve the manuscript. However, changes in the manuscript are at times rather brief and do not quite reflect the more extensive discussion in the comments & replies. I will only focus on two issues: the actual value of the stress exponent and the effect of a crystallographic preferred orientation (CPO).*

*The authors now do acknowledge (lines 35-44 in document with show changes) that although n=3 is normally used, actual observations do not always agree with it, referring to Cuffey & Kavanaugh (2011) and Budd & Jacka (1989). There are, of course, more papers (not cited) that suggest that n is unequal to 3, like n=4. The origin of n=3 is not really acknowledged: taking the minimum strain rate or maximum stress point, which is at only a few per cent of strain. This paper is about higher strains. Having added the few sentences, the manuscript continues as it was, i.e. effectively based on the assumption that n=3 is not only commonly used, but indeed correct. The suggestion from the original review to "include in*

*their following analysis what the consequences would be if n for natural flow is not 3, but perhaps indeed 4 as some claim to have measured in nature. Would this, for example, mean no contribution of GBS? Would the wattmeter work and give reasonable results?" is not considered.*

*On the contrary: In line 432, the authors write: " This provides an additional argument against applying the small grain growth exponents for bubble-free ice in the laboratory to natural settings. For example, if p = 2 the effective stress exponent for GBS-limited creep becomes 4.25. In this scenario, neither dislocation creep nor GBS-limited creep would result in an effective stress exponent that is consistent with the Glen law value." The authors argue thus that p should not be small, because the results would then not fit with n=3 of Glen's law, which is apparently taken as correct. If one acknowledges that n could be 4 (as I would say the actual velocity field of the Greenland ice sheet indicates), the conclusion would clearly be that p should be small, because it nicely fits the observations. This again shows that the growth law is a big uncertainty of the model.*

We agree that this sentence could be misleading, and have modified it accordingly on lines 399–409 of the revised text.

*A sceptical or malicious reader could interpret this as a circular argument: first fit the parameters to get n=3 and then claim success of the model, because it fits n=3. One could also come to the conclusion that if the model works for n=3 while ignoring mechanical anisotropy (see below) it must be wrong, since n=3 only applies to circa 3% strain and because ice is anisotropic. To avoid such unwanted interpretations, it would be good (or even imperative) if the authors would consider how well the wattmeter would work if n would be different, for example about 4.*

Using the composite flow law employed in our study, there are two ways to arrive at an effective stress exponent of ~4. The first is to have creep via grain size insensitive dislocation creep. The second is to have grain size sensitive GBS creep with a grain growth exponent $p = ~2$, consistent with bubble-free ice. The reason that we do not favor grain size sensitive creep with a grain growth exponent of 2 is that with these parameters the model cannot simultaneously fit the laboratory data and the ice core data. The figure below shows an application of the wattmeter using the grain growth parameters for Exp. 15 from Azuma et al. (2012) for bubble-free ice. Note that while this model under-predicts grain size in the laboratory experiment, it greatly overpredicts the grain size in the ice core. This is consistent with the extrapolation of the grain growth data shown in Figure 2 (blue curve corresponding to Exp. 15). Further tuning of the wattmeter parameters (e.g., $\lambda$) may improve the fit to one data set, but will degrade the fit to the other. We argue that this further supports the use of a higher grain growth exponent in the wattmeter, and that ice flow characterized by $n = 4$ mostly likely corresponds to dislocation creep.

We have described this result on lines 399–409 of the revised text. For the moment, we have chosen not to add this new figure to the manuscript. However, if either the reviewer or editor feels that it would be beneficial to include it in the final version, we would be happy to do so.

[Figure]

*The model focuses strongly (completely) on grain size as modifier of the effective viscosity. A big issue, raised in the review process, is actually the mechanical anisotropy of ice, which can greatly change the effective viscosity. After the review, the authors added only 2 short sentences (lines 522-524) addressing this elephant in the room. This I find rather meagre. What would be the effect of further grain size reduction that is mentioned? A lowering of the effective n for the whole ice sheet, if at the base where most shearing happens we may expect strong CPOs? Does that fit with some observations that grain size increases near the base? Or do other parameters need to be readjusted to refit the model to observations?*

This comment led us to go back and reassess our model for enhanced grain growth at the base of the ice sheet. Originally, we had noted that using the grain growth law for bubble-free ice ($p \sim 2$) resulted in basal grain sizes that were too large, and thus we settled on an intermediate $p$ value of 4 for basal ice in the pre-melting regime (Section 4.2). Following the reviewer's suggestion, we re-ran a model using the grain growth parameters for bubble-free ice, while simultaneously enhancing dislocation creep by a factor of 10 to simulate fabric development (based on the enhancement factor in Table 3.6 of Cuffey & Paterson, 2010). This does indeed result in a good fit to the grain sizes in the lowermost 200 m of ice (see dashed line in revised version of Figure 7). We have modified the text on lines 440–448 to describe these results.

Additional self-consistent modeling of fabric development coupled to grain size evolution is beyond the scope of the current study, but we do agree with the reviewer that this is an important avenue for future research.

*The wattmeter approach is of interest, as has already been proven for rocks other than ice, and therefore the paper could make a valuable contribution to glaciology. Without truly addressing alternatives to the isotropic Glen's law model, I am afraid it may be dismissed by those that acknowledge that that model is not realistic.*
*I therefore suggest publication after addressing these comments.*
*Kind regards, Paul Bons*